# LARGE LANGUAGE MODELS CAN SELF-IMPROVE AT WEB AGENT TASKS

## ABSTRACT

Training models to act as agents that can effectively navigate and perform actions in a complex environment, such as a web browser, has typically been challenging due to lack of training data. Large language models (LLMs) have recently demonstrated some capability to navigate novel environments as agents in a zero-shot or few-shot fashion, purely guided by natural language instructions as prompts. Recent research has also demonstrated LLMs have the capability to exceed their base performance through self-improvement, i.e. fine-tuning on data generated by the model itself. In this work, we explore the extent to which LLMs can self-improve their performance as agents in long-horizon tasks in a complex environment using the WebArena benchmark. In WebArena, an agent must autonomously navigate and perform actions on web pages to achieve a specified objective. We explore fine-tuning on three distinct synthetic training data mixtures and achieve a 31% improvement in task completion rate over the base model on the WebArena benchmark through a self-improvement procedure. We additionally contribute novel evaluation metrics for assessing the performance, robustness, capabilities, and quality of trajectories of our fine-tuned agent models to a greater degree than simple, aggregate-level benchmark scores currently used to measure self-improvement.

## 1 INTRODUCTION

Large language models (LLMs) have demonstrated impressive capabilities in a variety of natural language processing (NLP) tasks such as summarization and question answering (Radford et al., 2019; Raffel et al., 2020; Brown et al., 2020) through zero-shot and few-shot prompting techniques (Ouyang et al., 2022; Wei et al., 2021). However, prompting techniques alone are insufficient to enable LLMs to act as agents and navigate environments in order to solve complex, multi-step, long-horizon tasks (Yao et al., 2023). Fine-tuning LLMs to perform such tasks is also infeasible due to the scarcity of training data suitable for these tasks. Acquiring data for sequential decision-making and complex interactions is not only time-consuming, but also costly. Additionally, automatic evaluation of trajectories (or sequences of actions) taken by an agent is also difficult (Dinu et al., 2024). The absence of metrics that accurately capture the efficacy of each step in a sequence complicates the assessment of incremental improvements or degradations in an agent's performance.

A number of proposed self-improvement techniques have demonstrated that LLMs can use zero-shot and few-shot prompting to achieve performance above the baseline without any additional supervised training data (Huang et al., 2022; Chen et al., 2024). In place of supervised data as a learning signal, many of these techniques use a self-critique technique (Weng et al., 2022; Yuan et al., 2024), or obtain a critique through interactions with tools or environments (Gou et al., 2024). While self-improvement techniques have shown promise on standard NLP benchmark tasks like machine translation or question answering (Han et al., 2021; Huang et al., 2022; Chen et al., 2024), their efficacy has not yet been thoroughly investigated for long-horizon tasks that require multi-step interactions with a complex and realistic environment.

WebArena (Zhou et al., 2023) is a recently proposed benchmark wherein an LLM agent is required to solve tasks using a web browser. One example WebArena task is to use the OpenStreetMap website to answer the question "*What is the minimum travel time by car from CMU to University of Pittsburgh?*". Such a task requires an agent to complete a sequence of steps on the website, including entering a start location, entering a destination location, submitting a form, and then, reasoning

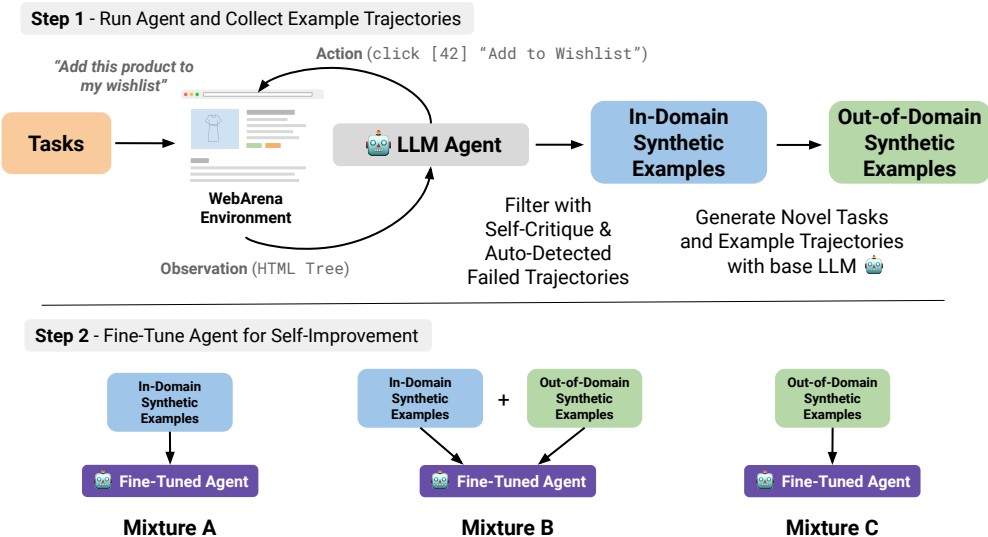

Figure 1: We generate synthetic data to fine-tune LLM agents to accomplish WebArena tasks such as "*Add this product to my wishlist*". **Step 1:** We first collect an initial set of trajectories, filter out low-quality trajectories in an unsupervised fashion, and keep the remainder as synthetic in-domain examples. We prompt our base LLM to generate novel out-of-domain tasks along with hypothetical solution trajectories by providing a few in-domain examples. **Step 2:** We then fine-tune our base LLM agent on each of the three distinct synthetic training data mixtures and evaluate performance.

over the result. The sequence of steps selected by an agent is called a *trajectory*. Unlike existing benchmarks, WebArena tasks are realistic and diverse, require dynamic interaction, and require navigating a complex environment. The baselines presented by Zhou et al. (2023) demonstrate that while LLMs are capable of interacting with this environment, even the strongest baseline, GPT-4 (OpenAI et al., 2024), is only able to solve ~14% of the tasks. This demonstrates that WebArena is a challenging benchmark even for the strongest frontier models (Chiang et al., 2024).

In summary, our contributions are:

- We propose and detail procedures for collecting and generating synthetic training examples for complex, multi-step tasks involving interaction with an environment. We explore collecting in-domain synthetic examples of trajectories as well as generating synthetic examples of solution trajectories for novel, out-of-domain tasks.

- We propose auxiliary metrics to understand the effect self-improvement has with respect to acquiring new capabilities and to evaluate variable-length trajectories produced by agents through an extension of the VERTEX score. These metrics provide nuanced insights not captured by aggregate-level benchmark scores currently used to evaluate self-improvement, allowing us to better assess the effect self-improvement has on multiple dimensions: performance, robustness, capability acquisition, and the quality of generated trajectories.

- We show that the performance of LLM agents improves after fine-tuning on this synthetic data, demonstrating that self-improving techniques work for a new class of tasks. We analyze three synthetic training data mixtures and find all three mixtures improve capability acquisition, with the best performing mixture yielding a 31% improvement over the base LLM agent on the WebArena benchmark. While we find models can self-improve once with our procedure, we find models can not iteratively self-improve.

## 2 SYNTHETIC DATA COLLECTION AND GENERATION

Self-improvement techniques for large language models typically involve using the model's own generations to create synthetic few-shot examples (Han et al., 2021) or synthetic fine-tuning data (Huang et al., 2022). These techniques amplify knowledge, correct behaviors, and introduce regularization

(Pham et al., 2022), often leading to an overall boost in performance. The self-generated examples are often filtered, post-edited, or ranked with a set of unsupervised techniques such as self-critique to introduce a signal for learning and improvement (Weng et al., 2022; Patel et al., 2023; Chen et al., 2024; Yuan et al., 2024). For multi-step agent tasks, the environment itself can additionally provide the LLM agent a way to detect failure in a fully unsupervised manner, which provides another useful signal for learning (Gou et al., 2024; Yuan et al., 2023; Song et al., 2024).

Using the WebArena benchmark (Zhou et al., 2023), we define and experiment with both in-domain synthetic training examples and out-of-domain synthetic training examples for web agent tasks, and fine-tune on three different synthetic data mixtures: **Mixture A** (in-domain synthetic examples only), **Mixture B** (both in-domain and out-of-domain synthetic examples), and **Mixture C** (out-of-domain synthetic examples only). Figure 1 illustrates our process.

**In-Domain Synthetic Data:** For all tasks in WebArena, we collect an initial set of trajectories using the base model. We filter out any trajectories where the model self-detected failure (self-critique) or failure was detectable in the environment and keep the remainder. We denote the remaining set of trajectories as *plausible trajectories*, where the model may or may not have completed the task successfully. Since lower-quality trajectories where the model outright failed to complete the task have been filtered out through self-detection, we hypothesize this remaining higher-quality set of plausible trajectories can serve as reasonably high-quality *in-domain synthetic examples* for fine-tuning. Similar to the self-improvement prior work we discuss earlier, the collection of this data is completely unsupervised and no ground-truth labels are utilized for filtering and selection.

**Out-of-Domain Synthetic Data:** We also evaluate whether the base model can generate completely novel tasks, objectives, web pages, and solution trajectories that can serve as useful training examples. We use the plausible trajectories as few-shot examples in a prompt for the base model to generate completely new tasks along with potential solution trajectories. To ensure the model generates examples with sufficient diversity and to improve generalization, we prompt the model to generate *out-of-domain synthetic examples* that are dissimilar from existing tasks and objectives as well as generate tasks for different websites than the set of 6 websites covered by the WebArena benchmark.

## 2.1 IN-DOMAIN SYNTHETIC DATA COLLECTION

The WebArena environment can be formulated as a partially observable Markov decision process: $\mathcal{E} = \langle \mathcal{S}, \mathcal{A}, \mathcal{O}, \mathcal{T} \rangle$, where $\mathcal{S}$ represents the state space, $\mathcal{A}$ represents the action space, $\mathcal{O}$ represents the observation space, and $\mathcal{T} : \mathcal{S} \times \mathcal{A} \to \mathcal{S}$ is the deterministic transition function (Zhou et al., 2023). An agent model $\mathcal{M}$ produces a next action $a_t \in \mathcal{A}$ provided an objective represented by some natural language intent $\mathbf{i}$, the current observation $o_t \in \mathcal{O}$, and the previous action taken $a_{t-1} \in \mathcal{A}$: $(\mathbf{i}, o_t, a_{t-1})$. This continues for $T$ time steps until the agent produces a stop action or the environment produces an error or stop condition. The models $\mathcal{M}$ we select for our experiments are the Qwen-1.5-72B-Chat model (Bai et al., 2023) and the Llama-3-70B-Instruct model (Dubey et al., 2024), which at the time of this work are highly ranked[1] and competitive open source LLMs (Chiang et al., 2024) that are accessible for fine-tuning. Further choice of inference parameters and other configuration details can be found in Appendix A.

Given this definition, we propose a procedure for sampling a set of in-domain synthetic training examples $\mathcal{D}_{\text{IN-DOMAIN}}$ where each training example is structured as $(\mathbf{i}, o_t, a_{t-1}) \to a_t$. These examples are sampled from a filtered set of trajectories collected by an initial run of the base agent model $\mathcal{M}$ over all tasks in WebArena:

---

[1] https://chat.lmsys.org/?leaderboard

---

**Algorithm 1** Collect In-Domain Synthetic Training Examples $\mathcal{D}_{\text{IN-DOMAIN}}$

---

**Input:** WebArena environment $\mathcal{E}$ and base agent model $\mathcal{M}$
**Output:** A set of in-domain synthetic training examples $\mathcal{D}_{\text{IN-DOMAIN}}$
  1: Initialize $\mathcal{P} \leftarrow \emptyset$        ▷ Set of plausible trajectories
  2: **for i** in WebArena benchmark **do**
  3:      Initialize trajectory $\mathcal{X} \leftarrow \emptyset$
  4:      Initialize observation $o_0 \leftarrow \text{INITIALOBSERVATION}(\mathcal{E}, \mathbf{i})$
  5:      Initialize action $a_{-1} \leftarrow \text{null}$
  6:      **for** $t = 0$ to $T$ **do**
  7:          $a_t \leftarrow \text{RUNAGENT}(\mathcal{M}, \mathbf{i}, o_t, a_{t-1})$
  8:          Append $(\mathbf{i}, o_t, a_{t-1}, a_t)$ to $\mathcal{X}$
  9:          **if** $a_t = \text{stop}$ **or** $\text{ENVIRONMENTERROR}(\mathcal{E}, a_t, o_{t+1})$ **then**
 10:              **break**
 11:          **end if**
 12:          $o_{t+1} \leftarrow \mathcal{T}(o_t, a_t)$        ▷ Observe updated state
 13:      **end for**
 14:      **if not** $\text{SELFCRITIQUE}(\mathcal{X})$ **and not** $\text{ISREFUSAL}(\mathcal{X})$ **and not** $\text{HASERROR}(\mathcal{X})$ **then**
 15:          Append $\mathcal{X}$ to $\mathcal{P}$        ▷ Filter out low-quality trajectories
                                                          to only keep plausible trajectories
 16:      **end if**
 17: **end for**
 18: Initialize $\mathcal{D}_i, \mathcal{D}_f, \mathcal{D}_{int} \leftarrow \emptyset$        ▷ Set of initial steps, final steps, intermediate steps
 19: **for** $\mathcal{X}$ in $\mathcal{P}$ **do**
 20:      Append $\mathcal{X}_0$ to $\mathcal{D}_i$
 21:      Append $\mathcal{X}_T$ to $\mathcal{D}_f$
 22:      **for** $t = 1$ to $T - 1$ **do**
 23:          Append $\mathcal{X}_t$ to $\mathcal{D}_{int}$
 24:      **end for**
 25: **end for**
 26: $\mathcal{D}_{\text{IN-DOMAIN}} \leftarrow \text{RANDSAMPLE}(\mathcal{D}_i, |\mathcal{D}_i|) \cup \text{RANDSAMPLE}(\mathcal{D}_f, |\mathcal{D}_i|) \cup \text{RANDSAMPLE}(\mathcal{D}_{int}, 2 * |\mathcal{D}_i|)$
 27: **return** $\mathcal{D}_{\text{IN-DOMAIN}}$

---

We filter out low-quality trajectories where the model produced a generation stating the task to be "impossible" or that it "cannot" make progress (a form of self-critique). Additionally, we filter out any trajectories where the model produced `stop[N/A]`, `stop[No ...]`, or `stop[]`, indicating when the model may have refused to provide an answer. Finally, we also filter out any trajectories where the WebArena environment encountered an error or the model failed to produce a valid, parsable generation. The final dataset of synthetic examples is balanced by randomly sampling an equal number of initial steps ($t = 0$), final steps ($t = T$), and intermediate steps ($t = 1 \ldots (T - 1)$) from the plausible trajectories in $\mathcal{P}$. In Table 1, we display how effective this unsupervised filtering process is by measuring the accuracy, precision, and recall of the 58 remaining trajectories kept in $\mathcal{P}$ from the 812 total trajectories to assess the proportion of correct/incorrect examples in $\mathcal{D}_{\text{IN-DOMAIN}}$.

| Set of Trajectories | # | Accuracy | F1 | Precision | Recall |
|---|---|---|---|---|---|
| All Trajectories | 812 | 0.071 | 0.133 | 0.071 | 1.000 |
| Plausible Trajectories $\mathcal{P}$ | 58 | 0.919 | 0.431 | 0.431 | 0.431 |

Table 1: Metrics on the proportion of trajectories that successfully completed the task in the set of plausible trajectories kept in $\mathcal{P}$ after filtering out low-quality trajectories. Approximately 43% of trajectories in $\mathcal{P}$ successfully completed the task, up from ~7% with no filtering, indicating useful learning signal is introduced by filtering using self-critiques and information from the environment.

## 2.2 OUT-OF-DOMAIN SYNTHETIC DATA GENERATION

Using examples from $\mathcal{D}_{\text{IN-DOMAIN}}$ as seed examples, we prompt our base LLM $\mathcal{M}$ to synthetically generate completely novel tasks, objectives, web pages, and solution trajectories to produce $\mathcal{D}_{\text{OUT-OF-DOMAIN}}$.

---

**Algorithm 2** Generate Out-of-Domain Synthetic Training Examples $\mathcal{D}_{\text{OUT-OF-DOMAIN}}$

---

**Input:** Base LLM model $\mathcal{M}$ and $\mathcal{D}_{\text{IN-DOMAIN}}$
**Output:** A set of out-of-domain synthetic training examples $\mathcal{D}_{\text{OUT-OF-DOMAIN}}$
1: Initialize $\mathcal{D}_{\text{OUT-OF-DOMAIN}} \leftarrow \emptyset$     ▷ Set of out-of-domain synthetic training examples
2: Initialize $\mathcal{I} \leftarrow \{\mathbf{i} \mid \mathbf{i} \in \text{WebArena benchmark}\}$     ▷ Set of 812 objectives in WebArena
3: Initialize $\mathcal{I}^* \leftarrow \emptyset$     ▷ Set of previously generated objectives
4: **for** $j = 1$ to $|\mathcal{D}_{\text{IN-DOMAIN}}|$ **do**
5:     **while** true **do**
6:         $\mathbf{i}^* \leftarrow \text{GENERATEOBJECTIVE}(\mathcal{M}, \text{RANDSAMPLE}(\mathcal{I}, 2) \cup \text{RANDSAMPLE}(\mathcal{I}^*, 2))$
7:         **if** $\max(\text{sim}(\mathbf{i}^*, \mathcal{I}^*)) < 0.70$ **then**     ▷ Ensure generated objectives are diverse
8:             Append $\mathbf{i}^*$ to $\mathcal{I}^*$
9:             **break**
10:         **end if**
11:     **end while**
12:     $\mathbf{p}^* \leftarrow \text{GENERATEPLAN}(\mathcal{M}, \mathbf{i}^*)$     ▷ Generate an outline of a hypothetical solution trajectory
13:     $k \leftarrow \text{RANDCHOICE}(\{1, \ldots, |\mathbf{p}^*|\})$     ▷ Randomly select one of the steps in the plan, weighted to equally balance initial, final, and intermediate steps
14:     $a_{t-1}^*, a_t^* \leftarrow \text{GENERATEACTIONS}(\mathcal{M}, \text{RANDSAMPLE}(\mathcal{D}_{\text{IN-DOMAIN}}, 2), \mathbf{i}^*, \mathbf{p}^*, k)$
15:     $o_t^* \leftarrow \text{GENERATEOBSERVATION}(\mathcal{M}, \text{RANDSAMPLE}(\mathcal{D}_{\text{IN-DOMAIN}}, 2), \mathbf{i}^*, \mathbf{p}^*, k)$
16:     Append $(\mathbf{i}^*, o_t^*, a_{t-1}^*, a_t^*)$ to $\mathcal{D}_{\text{OUT-OF-DOMAIN}}$
17: **end for**
18: **return** $\mathcal{D}_{\text{OUT-OF-DOMAIN}}$

---

When generating new objectives, we use 4 few-shot examples (two objectives sampled from tasks in WebArena and two sampled from previously generated objectives). We use 2 few-shot examples when generating previous actions, next actions, and observations (web pages in the form of accessibility trees). We use a temperature of 1.0 and set top-p to 1.0 during generation. Detailed information on the prompts used for generating $\mathcal{D}_{\text{OUT-OF-DOMAIN}}$ can be found in Appendix G. When generating novel objectives, we specifically prompt the model to generate objectives that are dissimilar to the example objectives to encourage out-of-domain generations. We also ensure each novel objective has $< 0.70$ cosine similarity with any objective previously generated using the `all-distilroberta-v1` sentence similarity model (Reimers and Gurevych, 2019; Liu et al., 2019; Sanh et al., 2019) to promote diversity. Table 2 gives examples of out-of-domain objectives that our method generated.

| Objectives in WebArena Benchmark | Generated Out-of-Domain Objectives |
|---|---|
| • Tell me the total cost of my latest pending order? (Shopping)
• Compare the time for walking and driving route from AMC Waterfront to Univ of Pittsburgh (Maps)
• Check out the most recent open issues (GitLab)
• Which customer has placed 2 orders in the entire history? (Shopping Admin)
• ... | • Locate and purchase a subscription to The Economist digital edition (`https://store.economist.com/...`)
• Find the nutrition facts for a Grilled Chicken Caesar Salad from Chili's (`http://www.chilis.com/...`)
• Find the active coupons for a one-year subscription to Adobe Creative Cloud (`https://www.couponcabin.com/...`)
• Subscribe to the premium plan for Grammarly to unlock advanced writing features. (`https://www.grammarly.com/...`)
• ... |

Table 2: A sample of the novel objectives generated compared with the objectives found in WebArena. A full sample of a generated out-of-domain synthetic example can be found in Appendix B.

## 3 EVALUATION

We perform evaluation using the standard metrics proposed by the WebArena benchmark like functional correctness (Zhou et al., 2023) as well as evaluate with new auxiliary metrics we propose that give more nuanced insight into an agent's performance.

### 3.1 FUNCTIONAL CORRECTNESS SCORE

Functional correctness is the standard metric proposed by the WebArena benchmark that is a simple binary task completion score (0 or 1) averaged over all 812 tasks in the benchmark.

### 3.2 CAPABILITY SCORE (NEW)

While WebArena contains 812 unique task instances, these 812 tasks are instantiated using natural language intent templates like "What is the minimum travel time by car from {{location1}} to {{location2}}?". Therefore, many tasks actually test the same *capability*. Aggregate-level metrics like the functional correctness score may be misleading since improvements may only be due to the model becoming more robust at solving capabilities it already could solve versus demonstrating the ability to solve new capabilities that were previously unsolvable. There are 241 unique templates in WebArena that are used to instantiate 812 tasks. Moreover, some of these templates are simple paraphrases of each other. For example, "What is the estimated driving time between {{city1}} and {{city2}}?" is a paraphrase of the prior template. Using a sentence similarity model,[2] we iteratively group these templates into a set of unique capabilities. Each template is grouped with any existing capability if it has a similarity of $> 0.60$ with any template in the group, otherwise the template is added to a new capability group. This results in 136 unique capabilities (see Appendix F). A model receives a score of 1 for each capability group with at least one successful task completed, otherwise it receives a score of 0.[3] The capability score is then the averaged over all 136 capabilities.

We note, however, that a number of tasks in the WebArena benchmark are trivial tasks and can be solved by a trivial baseline agent or weak model that performs no actions and only immediately exits by always generating stop [N/A]. In the capability score computation, we do not count such trivial tasks as evidence a model can perform the capability as these are degenerate cases of the capability.

### 3.3 VERTEX$_{\text{DTW}}$ SCORE (NEW)

Both functional correctness and the capability score only evaluate task completion, however, they do not assess the quality of entire trajectories, therefore, a measure that is sensitive to incremental improvements and degradations in trajectories, independent of task completion, is desirable. We extend the recently proposed VERTEX score (Dinu et al., 2024), which measures the similarity of two relational trajectories by using embeddings to compare node distributions within a computational graph. The VERTEX score integrates the semantic meaning across the distributional path by computing at each node the cross-similarity between the generated embeddings and embeddings sampled from a reference distribution. An ideal reference distribution would be ground-truth reference trajectories produced by humans for all of the WebArena tasks. In absence of this, we use a larger, stronger model, GPT-4 (OpenAI et al., 2024), to collect three reference trajectories for each task.

One obstacle to the straightforward application of the VERTEX score is the assumption that both trajectories are of the same length. Agents operating in complex environments, however, are not constrained to a fixed-length for the trajectories they produce. Therefore, we propose modification in the computation of the VERTEX score that enables comparison of sequences with different lengths. Our extension consists of an additional alignment step prior to calculating the VERTEX score for the aligned trajectories. First, we embed all steps of a trajectory $\mathcal{X}$ as $e_t = f(o_t, a_t) \in R^d$, where $f$ is an embedding model[4] with embedding dimension $d$. The embedding model $f$ is independent of both the model that generated the reference trajectories as well as the model that generated the test trajectories. Then, we use *Dynamic Time Warping* (DTW) (Berndt and Clifford, 1994) to align two embedded trajectories $\tilde{\mathcal{X}}_m = (e_0, \ldots, e_i, \ldots, e_m) \in R^{m \times d}$ and $\tilde{\mathcal{X}}_n = (e_0, \ldots, e_j, \ldots, e_n) \in R^{n \times d}$ with length $m$ and $n$, respectively. Consequently, we refer to our proposed measure as VERTEX$_{\text{DTW}}$. DTW returns an alignment path $\nu$ of length $T$, where each $e_i \in \tilde{\mathcal{X}}_m$ is aligned with a corresponding $e_j \in \tilde{\mathcal{X}}_n$, preserving the order in their respective trajectory. This order preservation occurs because once a node is matched, it is excluded from potential new matches, maintaining the integrity of the temporal alignment. As a scoring function for DTW, we choose cosine distance. In addition to the

---

[2]We use the all-distilroberta-v1 sentence similarity model (Sanh et al., 2019).

[3]Since we do not count trivial tasks as a successful completion, a single successful completion of a capability provides sufficient evidence of acquisition. We discuss robustness and consistency separately in Section 5.

[4]We use the all-mpnet-base-v2 embedding model (Song et al., 2020).

alignment step, we introduce a linear distance decay factor that decreases the contribution of aligned embeddings if they are far apart in the original trajectories. Once two trajectories are aligned, we compute the VERTEX score by Eq. (4) in Dinu et al. (2024) with the addition of the distance decay. Therefore, the VERTEX$_{\text{DTW}}$ score is computed as:

$$s(\tilde{\mathcal{X}}_{\text{ref}}, \tilde{\mathcal{X}}_{\text{test}}, \nu) := \frac{1}{T} \int_{t_0}^{t_T} \left[ \min(\max(0, \frac{1}{1 + |i_{\nu_t} - j_{\nu_t}|} \widetilde{\text{MMD}}^2(e_{\text{ref}}^{\nu_t}, e_{\text{test}}^{\nu_t}) - z_{\text{rand}}), 1) \right] dt, \quad (1)$$

where $i_{\nu_t}$ and $j_{\nu_t}$ are the position indices in the alignment path $\nu$ at time $t$, $\tilde{\mathcal{X}}_{\text{ref}}$ and $\tilde{\mathcal{X}}_{\text{test}}$ are aligned trajectories of embeddings from the reference set and the model under test, respectively, and $z_{\text{rand}}$ is a baseline correction from a random baseline.[5] Furthermore, if we have multiple reference sequences for a given task, we compute the VERTEX$_{\text{DTW}}$ score for every reference sequence and choose the maximum score, under the assumption that they describe different paths for solving the task.

## 4 EXPERIMENTS

We perform a number of experiments fine-tuning agent models on the synthetic training data mixtures we discuss in Section 2 and assess the extent to which the agent model has self-improved over base agent model $\mathcal{M}$ with our evaluation metrics. Table 3 displays the results of these experiments.

### 4.1 BASELINE AGENT PERFORMANCE

As baselines, we evaluate our base agent model $\mathcal{M}$ as well as implement a trivial agent that always outputs stop [N/A]. A number of tasks in WebArena can be solved by this trivially implementable agent or a weak model that always refuses to continue and exits immediately, therefore, our trivial agent baseline helps discriminate which tasks being completed successfully should contribute to an agent being meaningfully capable when computing the capability score.

### 4.2 SELF-IMPROVEMENT FINE-TUNED AGENT PERFORMANCE

We fine-tune our base agent model $\mathcal{M}$ on the 3 synthetic dataset mixtures previously discussed: 1) $\mathcal{D}_A = \mathcal{D}_{\text{IN-DOMAIN}}$ 2) $\mathcal{D}_B = \mathcal{D}_{\text{IN-DOMAIN}} \cup \mathcal{D}_{\text{OUT-OF-DOMAIN}}$ and 3) $\mathcal{D}_C = \mathcal{D}_{\text{OUT-OF-DOMAIN}}$ with a straightforward auto-regressive loss using QLoRA (Dettmers et al., 2023; Hu et al., 2021):

$$L_{\text{FT}}(\theta) = -\mathbb{E}_{[(\mathbf{i}, o_t, a_{t-1}), a_t] \sim \mathcal{D}} \left[ \log P_\theta(a_t \mid (\mathbf{i}, o_t, a_{t-1})) \right]$$

to produce $\mathcal{M}_A$, $\mathcal{M}_B$, and $\mathcal{M}_C$. We perform a 90/10% train-validation split of our datasets and train with an early stopping patience of 5 epochs, using a batch size of 16 examples and a learning rate of 1e-5. Further details about training configuration and hyperparameters can be found in Appendix A.

### 4.3 ITERATIVE SELF-IMPROVEMENT FINE-TUNED AGENT PERFORMANCE

We also experiment with iterative self-improvement (Chen et al., 2024) to assess whether further improvement can be gained from a subsequent round of our self-improvement procedure. We perform this experiment on Mixture A. It is conceivable that after fine-tuning on $\mathcal{D}_A^1$, filtering from a set of trajectories with higher performance might yield a stronger set of plausible trajectories[6] to produce $\mathcal{D}_A^2$. Mixtures B and C are less likely to demonstrate improvement over a subsequent round since the fine-tuned models are not specifically trained to generate better synthetic out-of-domain examples.

## 5 DISCUSSION

We summarize key results from our experiments as well as discuss insights towards the efficacy of our self-improvement procedures for complex, multi-step tasks like web agent tasks.

---

[5]We use the trivial agent implementation described in Section 4.1 for baseline correction in our computation.

[6]To maximize data for iterative self-improvement, during filtering, we also fallback to checking the base model trajectory for a task if the self-improved model's trajectory for a task is filtered out.

| Agent Model | Functional Correctness | Capability | VERTEX$_{DTW}$ |
|---|---|---|---|
| *Baseline Agents* | | | |
| Trivial Agent | 4.68 | 0.00 | - |
| Base Agent Model ($\mathcal{M}$) | 7.14 | 15.44 | 0.35 |
| *Qwen-72B Self-Improvement* | | | |
| Base Agent Model ($\mathcal{M}$) | 7.2 ± 0.17 | 11.62 ± 0.28 | 0.35 ± 0.003 |
| Agent Model Fine-Tuned on Mixture A ($\mathcal{M}_A$) | 8.81 ± 0.2 | **14.19 ± 0.35** | **0.38 ± 0.003** |
| Agent Model Fine-Tuned on Mixture B ($\mathcal{M}_B$) | **9.57 ± 0.22** | **14.54 ± 0.3** | 0.35 ± 0.0029 |
| Agent Model Fine-Tuned on Mixture C ($\mathcal{M}_C$) | 6.11 ± 0.2 | **12.4 ± 0.3** | 0.28 ± 0.0027 |
| *Llama-3-70B Self-Improvement* | | | |
| Base Agent Model ($\mathcal{M}$) | 6.6 ± 0.16 | 12.96 ± 0.29 | **0.28 ± 0.0027** |
| Agent Model Fine-Tuned on Mixture A ($\mathcal{M}_A$) | **7.38 ± 0.2** | **15.08 ± 0.3** | **0.28 ± 0.0028** |
| Agent Model Fine-Tuned on Mixture B ($\mathcal{M}_B$) | 6.67 ± 0.17 | **14.64 ± 0.3** | 0.26 ± 0.0032 |
| Agent Model Fine-Tuned on Mixture C ($\mathcal{M}_C$) | 6.3 ± 0.18 | **13.68 ± 0.3** | 0.27 ± 0.0031 |
| *Iterative Self-Improved Agents* | | | |
| Agent Model 2x Fine-Tuned on Mixture A ($\mathcal{M}_A^2$) | 8.37 | 16.91 | 0.37 |

Table 3: Evaluation metrics on WebArena for baseline agents and self-improved agent models over two models. We include confidence intervals and bold significant (p=0.05) values (Efron, 1979) that outperform the baseline (self-improve).

**Can models self-improve at web agent tasks?** We find fine-tuning on both Mixtures A and B improve overall benchmark performance with the best performing mixture, Mixture B, completing 18 more tasks correctly, a 31% relative improvement (7.14 → 9.36). Training on all Mixtures A, B, and C demonstrate self-improvement on at least one metric, with $\mathcal{M}_C$ showing a gain on capability score.

**Do self-improved agents acquire new capabilities?** We find agent models can acquire new capabilities through self-improvement, however, they also may lose the ability to perform some capabilities. In net, all of our self-improved agents acquire more capabilities than they lose. We find fine-tuning on both Mixtures A and B improve the capability score equally and lead to the largest net acquisition of capabilities demonstrating 5 more capabilities than the base agent model, a 24% relative improvement (15.44 → 19.12). We find all agent models demonstrate at least one new capability that no other agent model demonstrates, for example, only $\mathcal{M}_C$ successfully completes the "Fork {{repo}}" capability on the GitLab website. Interestingly, we find that the majority of capabilities acquired by $\mathcal{M}_A$ and $\mathcal{M}_C$ are mutually exclusive, suggesting in-domain synthetic examples and out-of-domain synthetic examples improve acquisition of different capabilities. We list all capabilities $\mathcal{M}_A$, $\mathcal{M}_B$, and $\mathcal{M}_C$ acquire and lose compared to $\mathcal{M}$ in Appendix C.

**Are self-improved agents more robust?** For $\mathcal{M}_B$, we find a larger improvement in functional correctness (31%) than in capability score (24%), which supports that the agent model is improving at more consistently succeeding at tasks belonging to the same capability, an indicator of one type of robustness. $\mathcal{M}_C$ is less robust by the same measure. Moreover, the capability analysis in Appendix C also shows both $\mathcal{M}_A$ and $\mathcal{M}_B$ after self-improvement still demonstrate the majority of capabilities demonstrated by the base agent model $\mathcal{M}$, whereas $\mathcal{M}_C$ only demonstrates a minority. This would indicate $\mathcal{M}_A$ and $\mathcal{M}_B$ more reliably maintain the capabilities of the base agent model after self-improvement, a measure of robustness that would be useful in deployed settings where users of agent models may desire stability in performance.

**Is there an effect on the quality of generated trajectories?** Fine-tuning on Mixtures A and B show no degradation in the quality of generated trajectories and show small improvement towards the reference on VERTEX$_{DTW}$. Fine-tuning on Mixture C degrades the the quality of generated trajectories from the reference. Training on the out-of-domain synthetic examples allows $\mathcal{M}_C$ to demonstrate some unique capabilities no other agent model demonstrates, however, inspecting trajectories from $\mathcal{M}_C$, we find this comes with trade-offs. For example, compared with $\mathcal{M}_A$, we find $\mathcal{M}_C$ produces longer trajectories (~1.6x) and produces more invalid actions (~3.9x). In comparison with $\mathcal{M}$, $\mathcal{M}_A$ and $\mathcal{M}_B$ do not greatly increase trajectory length (~1.1x and ~1.3x) or the rate of invalid actions (~1x and ~1.3x), further explaining the quality difference VERTEX$_{DTW}$ highlights.

Due to lack of human reference, the reliability of this evaluation is limited which we discuss in Section 7. In Appendix D, we compute variants of VERTEX$_{\text{DTW}}$, weighting by capability and filtering out trivial tasks. We find these variants make little difference in the relative ranking of agent models.

**Can models iteratively self-improve at web agent tasks?**  Our results are consistent with prior works such as Chen et al. (2024) and Feng et al. (2024) and we find diminishing returns to successive rounds of self-improvement and training on synthetic data. While the agent model after a second round of self-improvement outperforms the base agent model, it does not perform any better than agent models with a single round of self-improvement. We analyze the set of plausible trajectories in the second round in Appendix E and find that while more synthetic training examples can be collected, they are of lower quality and contain a higher proportion of failed trajectories.

## 6 RELATED WORK

**Self-Improvement**  A number of techniques have been proposed for self-improving LLMs (Huang et al., 2022; Weng et al., 2022; Madaan et al., 2023, *inter alia*). Some self-improvement techniques (Han et al., 2021; Gulcehre et al., 2023; Singh et al., 2024; Chen et al., 2024; Yuan et al., 2024) involve self-distillation (Zhang et al., 2019), a special form of knowledge distillation (Hinton et al., 2015) where the teacher and student are the same model. A growing trend of works (Wang et al., 2023; Gunasekar et al., 2023) similarly prompt LLMs to generate synthetic fine-tuning data.

**LLM Agents**  A number of prompting techniques proposed (Kojima et al., 2023; Wei et al., 2022; Yao et al., 2023; Shinn et al., 2023) can improve an LLM agent's performance, however, these techniques are orthogonal to self-improvement fine-tuning. Chen et al. (2023) introduces a technique for supervised fine-tuning of LLM agents. Sodhi et al. (2024) and Lai et al. (2024) introduce handcrafted subprompts or supervised techniques that improve performance on WebArena.

**Self-Improving Agents**  Bousmalis et al. (2023) demonstrates self-improving embodied agents for complex robotics tasks. Aksitov et al. (2024) introduces a method for self-improving agents on a simpler multi-step question answering task. Concurrently, Song et al. (2024) proposes a similar procedure of filtering trajectories and fine-tuning, but primarily focuses on supervised filtering, does not explore generating novel tasks and synthetic data, and evaluates on less realistic and complex benchmarks. Pan et al. (2024) explores using vision models for critique to improve on WebArena.

## 7 LIMITATIONS AND BROADER IMPACTS

While we find self-improvement fine-tuning techniques can improve performance by reinforcing correct actions and decisions of an underlying model, these techniques can also further reinforce incorrect actions and biases of the underlying model. Some human or supervised filtering may mitigate this drawback, however, in this paper we focus our investigation on the efficacy and quality of unsupervised self-improvement as producing datasets for such complex tasks is difficult and expensive. Our analysis of capabilities is limited by our method to group tasks by the intent template used and cosine similarity. It is possible other strategies may produce more optimal groups to measure capabilities. Our VERTEX$_{\text{DTW}}$ score utilizes a stronger model's generations (GPT-4) as a reference, however, human references would significantly improve the reliability of this evaluation. While WebArena spans many different types of realistic tasks and websites (shopping, online forums, maps, etc.), a future direction for this work might involve evaluation on larger, and more diverse benchmark. Some hyperparameter choices and parameter choices are chosen arbitrarily, within the limits of what was computationally feasible during our experiments and future work may seek to explore the sensitivity of such techniques to hyperparameters.

## 8 CONCLUSION

In this work, we explore whether large language models can self-improve beyond their base performance at complex, long-horizon web agent tasks. We conclude self-improvement can increase the performance and robustness of agent models and allow agent models to acquire new capabilities.

We also find it is possible for self-improvement to yield these benefits with minimal degradation to the quality of trajectories. The self-improvement procedures we propose are a promising step towards boosting the performance of LLMs in complex, multi-step agent environments such as web environments, without relying on supervised training data. We release our code, evaluation metrics with references, synthetic datasets, and model trajectories.

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

APPENDIX

## A  TRAINING AND INFERENCE DETAILS

| Hyperparameter | Value |
|---|---|
| Model | Qwen/Qwen1.5-72B-Chat |
| Hardware | 2x NVIDIA RTX A6000 |
| Distributed Protocol | PyTorch FSDP |
| Data Type | `torch.bfloat16` |
| Quantization | 4-bit (nf4), double quantized |
| LoRA | `all-linear, r=8`
`lora_alpha=8`
`lora_dropout=0.0` |
| Optimizer | `adamw_torch` |
| Learning Rate | 1e-5 |
| Weight Decay | 0.01 |
| Learning Rate Scheduler | `linear` |
| Warmup Steps | 0 |
| Batch Size | 16 |
| Train-Validation Split | 90/10% |
| Early Stopping Threshold | 0.0 |
| Early Stopping Patience | 5 epochs |

Table 4: Hyperparameters selected for fine-tuning experiments.

| Inference Parameter | Value |
|---|---|
| Model | Qwen/Qwen1.5-72B-Chat |
| Hardware | 4x NVIDIA RTX A6000 |
| Data Type | `torch.bfloat16` |
| Quantization | 4-bit (nf4), double quantized |
| Prompt Template | `p_cot_id_actree_2s` |
| Temperature | 1.0 |
| Top-P | 0.9 |
| Max New Tokens | 384 |

Table 5: Parameters used during inference, we follow the default parameters for inferencing set by the WebArena benchmark (Zhou et al., 2023).

## B  SAMPLE OF GENERATED OUT-OF-DOMAIN SYNTHETIC TRAINING EXAMPLE

| | |
|---|---|
| **Objective** | Subscribe to the premium plan for Grammarly to unlock advanced writing features. |
| **URL** | `https://www.grammarly.com/plans` |
| **Plan** | 1. Click on the "Go Premium" button located at the top right corner of the page.
2. Scroll down to review the available plans (if needed).
3. Click on the "Choose Plan" button for the preferred premium plan.
4. Provide payment information, such as credit card details, in the respective input fields.
5. Enter your billing information, including name, address, and zip code (if required).
6. Type your email address again to confirm it.
7. Create a password for your Grammarly account and type it into the password field.
8. Check the box to agree to the terms and conditions.
9. Click on the "Start Your Free Trial" (or similar) button if you qualify for a trial.
10. Review the confirmation message and wait for the subscription activation.
11. Stop when the subscription is successfully activated and the premium features are unlocked. |
| **Selected Step in Plan** | 3 |
| **Previous Action** | click [1098] where [1098] is Generic 'Premium' |
| **Next Action** | Let's think step-by-step. To proceed with subscribing to the premium plan, the user must now choose their plan. They have already clicked on the 'Premium' plan option. The next logical action is to click on the 'Choose Plan' button to finalize the selection and proceed to the payment page. In summary, the next action I will perform is ```click [1105]``` to select the premium plan and move to the next step in the subscription process. |
| **Observation** | [1] RootWebArea 'Grammarly plans - Premium, Business & Student Writing Tools' focused: True
[125] Generic 'Grammarly plans - Premium, Business & Student Writing Tools'
    [127] link 'Privacy Policy'
    [130] link 'Terms of Use'
    [133] link 'Cookies'
    [136] link 'TrustArc Privacy Notice'
    [587] link 'Language'
    [1054] link 'United States English'
    [592] link 'Logout'
    [595] link 'Invite'
    [598] link 'Get the Extension'
    [128] link 'Login'
    [601] link 'Sign Up'
    [580] Generic 'Plans'
        [583] div 'Free'
            [586] link 'Get Started'
        [590] Generic 'Premium'
            [1098] Generic 'Premium' haspopup: submenu expanded: False
                [640] link 'View Features'
                [643] StaticText 'Start Now!'
                [1105] button 'Choose Plan'
        [1089] Generic 'Business'
            [649] link 'View Pricing'
            [652] StaticText 'Get Quote'
        [1092] Generic 'Student'
            [646] link 'View Pricing'
            [649] StaticText 'Start Now!'
    [566] Generic 'Write with confidence, wherever you work'
        [570] StaticText 'Powerful writing tools for work, school, and everything in between.'
        [574] button 'Learn More & Try it Free'
    [1052] table '
        [569] row '
            [574] rowheader 'Free'
            [575] gridcell '
        [568] row '
            [573] rowheader 'Premium'
            [574] gridcell '
        [567] row '
            [572] rowheader 'Business'
            [573] gridcell '
        [566] row '
            [571] rowheader 'Student'
            [572] gridcell ' |

Table 6: A selected full sample of a generated novel, out-of-domain synthetic training example in $\mathcal{D}_{\text{OUT-OF-DOMAIN}}$.

## C CAPABILITY ANALYSIS

| Agent Model | Net Change | Capabilities Acquired | Capabilities Lost |
|---|---|---|---|
| $\mathcal{M}_A$ | +5 | 1. Tell me the count of comments that have received more downvotes than upvotes for the user who made the latest post on the `{{forum}}` forum.
2. Find a subreddit focused on topics related to `{{topic}}`, and post my question, "`{{question}}`" there
3. Measure distance between `{{location/address_1}}` and `{{location/address_2}}` by walking
4. Tell me the coordinates of `{{location}}` in DD format
5. Show me the "`{{product}}`" listings by `{{sorting_order}}`.
6. Open my latest updated issue that has keyword "`{{keyword}}`" in its title to check if it is closed
7. Reply to `{{position_description}}` with my comment "`{{content_description}}`"
8. Tell me the distance to drive from Carnegie Mellon University to the top computer science school in massachusetts
9. How many commits did `{{user}}` make on `{{date}}` in total? | 1. Tell me the total cost of my latest `{{status}}` order?
2. Checkout merge requests assigned to me
3. Today is 6/12/2023. Tell me how many fulfilled orders I have `{{period}}`, and the total amount of money I spent.
4. Subscribe to the newsletter of OneStopMarket |
| $\mathcal{M}_B$ | +5 | 1. Tell me the count of comments that have received more downvotes than upvotes for the user who made the latest post on the `{{forum}}` forum.
2. What is the minimum travel time by car from `{{location1}}` to `{{location2}}`?
3. Find a subreddit focused on topics related to `{{topic}}`, and post my question, "`{{question}}`" there
4. See all public projects
5. Set my gitlab status as `{{status}}`.
6. Show me the route and driving time from `{{city1}}` to `{{city2}}`
7. Ask for advice about `{{issue}}` in a subreddit for relations
8. Show me the "`{{product}}`" listings by `{{sorting_order}}`.
9. Reply to `{{position_description}}` with my comment "`{{content_description}}`"
10. Show me the way from `{{location}}` to the home stadium of `{{sport_team}}` `{{time}}` | 1. Checkout merge requests assigned to me
2. Today is 6/12/2023. Tell me how many fulfilled orders I have `{{period}}`, and the total amount of money I spent.
3. Subscribe to the newsletter of OneStopMarket
4. Show me the command to clone `{{repo}}` with SSH.
5. Show me the `{{info}}` for order number `{{order_number}}`. |

*Continued on next page...*

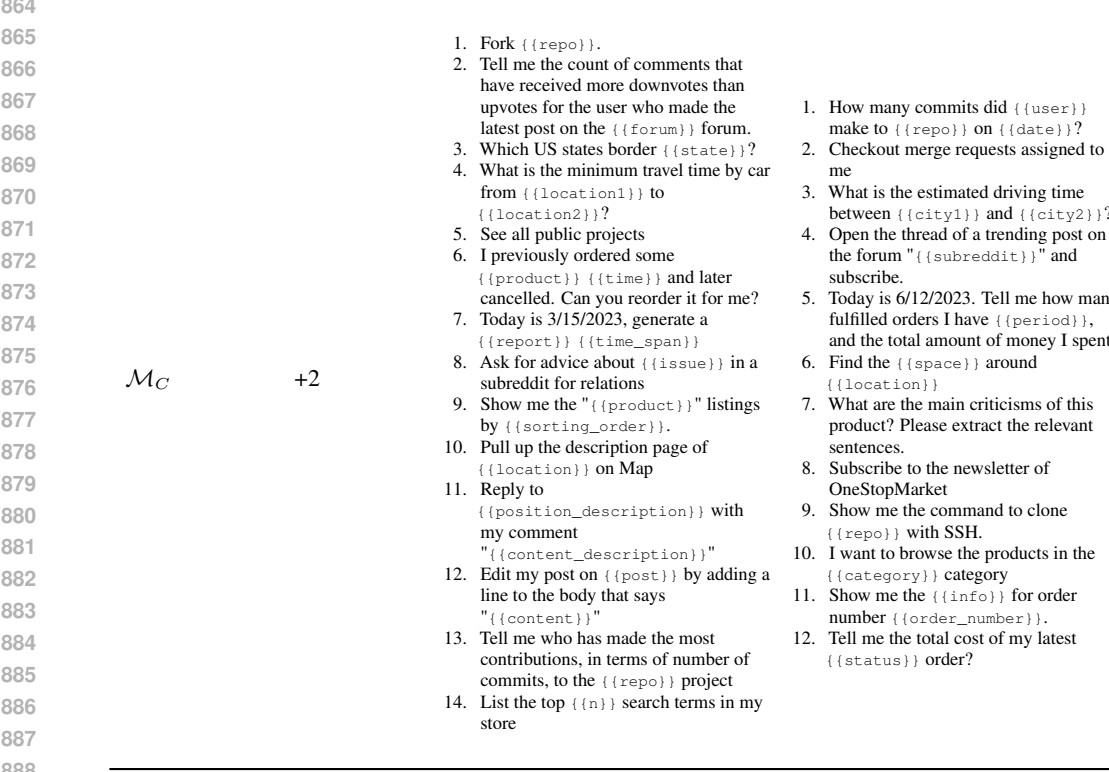

$\mathcal{M}_C$      +2

1. Fork `{{repo}}`.
2. Tell me the count of comments that have received more downvotes than upvotes for the user who made the latest post on the `{{forum}}` forum.
3. Which US states border `{{state}}`?
4. What is the minimum travel time by car from `{{location1}}` to `{{location2}}`?
5. See all public projects
6. I previously ordered some `{{product}}` `{{time}}` and later cancelled. Can you reorder it for me?
7. Today is 3/15/2023, generate a `{{report}}` `{{time_span}}`
8. Ask for advice about `{{issue}}` in a subreddit for relations
9. Show me the "`{{product}}`" listings by `{{sorting_order}}`.
10. Pull up the description page of `{{location}}` on Map
11. Reply to `{{position_description}}` with my comment "`{{content_description}}`"
12. Edit my post on `{{post}}` by adding a line to the body that says "`{{content}}`"
13. Tell me who has made the most contributions, in terms of number of commits, to the `{{repo}}` project
14. List the top `{{n}}` search terms in my store

1. How many commits did `{{user}}` make to `{{repo}}` on `{{date}}`?
2. Checkout merge requests assigned to me
3. What is the estimated driving time between `{{city1}}` and `{{city2}}`?
4. Open the thread of a trending post on the forum "`{{subreddit}}`" and subscribe.
5. Today is 6/12/2023. Tell me how many fulfilled orders I have `{{period}}`, and the total amount of money I spent.
6. Find the `{{space}}` around `{{location}}`
7. What are the main criticisms of this product? Please extract the relevant sentences.
8. Subscribe to the newsletter of OneStopMarket
9. Show me the command to clone `{{repo}}` with SSH.
10. I want to browse the products in the `{{category}}` category
11. Show me the `{{info}}` for order number `{{order_number}}`.
12. Tell me the total cost of my latest `{{status}}` order?

Table 7: Capabilities acquired and lost compared to the base agent model $\mathcal{M}$, along with the net change in the total number of capabilities demonstrated, for each self-improved fine-tuned agent model.

# D  FULL VERTEX$_{\text{DTW}}$ SCORE RESULTS

| Agent Model | VERTEX$_{\text{DTW}}$ | VERTEX$_{\text{DTW-bycap}}$ | VERTEX$_{\text{DTW-notrivial}}$ |
|---|---|---|---|
| *Baseline Agents* | | | |
| Base Agent Model ($\mathcal{M}$) | 0.35 | 0.40 | 0.38 |
| *Self-Improved Agents* | | | |
| Agent Model Fine-Tuned on Mixture A ($\mathcal{M}_A$) | **0.38** | **0.42** | 0.42 |
| Agent Model Fine-Tuned on Mixture B ($\mathcal{M}_B$) | 0.35 | 0.40 | 0.40 |
| Agent Model Fine-Tuned on Mixture C ($\mathcal{M}_C$) | 0.28 | 0.33 | 0.34 |
| *Iterative Self-Improved Agents* | | | |
| Agent Model 2x Fine-Tuned on Mixture A ($\mathcal{M}_A^2$) | 0.37 | 0.41 | **0.43** |

Table 8: Variants of the VERTEX$_{\text{DTW}}$ score metric: 1) computed over all trajectories 2) weighting the trajectories by capability 3) weighting the trajectories by capability and filtering out trajectories for trivial tasks.

# E ITERATIVE SELF-IMPROVEMENT PLAUSIBLE TRAJECTORIES

| Set of Trajectories | # | Accuracy | F1 | Precision | Recall |
|---|---|---|---|---|---|
| All Trajectories | 812 | 0.071 | 0.133 | 0.071 | 1.000 |
| Plausible Trajectories $\mathcal{P}^1$ | 58 | **0.919** | **0.431** | **0.431** | **0.431** |
| Plausible Trajectories $\mathcal{P}^2$ | **131** | 0.825 | 0.317 | 0.252 | 0.429 |

Table 9: Metrics on the proportion of trajectories that successfully completed the task in the set of plausible trajectories kept in $\mathcal{P}$ after filtering out low-quality trajectories for each iterative round of self-improvement. On the second round of self-improvement, we keep 131 plausible trajectories making our potential synthetic training dataset larger, however, the accuracy and P/R/F1 metrics indicate it would be a lower quality dataset to fine-tune on.

# F    CAPABILITIES IN WEBARENA

In this appendix, we list the grouping of tasks into "capabilities" we find in WebArena using the automated method we describe in Section 3.2. These tasks are grouped by the intent template used by WebArena to create the task as well as cosine similarity to group paraphrases detected by a sentence similarity model. We do not perform manual modifications to the groups and instead solely rely on automated techniques. We acknowledge grouping of natural language task objectives into capability areas is subjective and discuss this a limitation in Section 7:

**Capability #1:**
- What are the top-`{{n}}` best-selling product in `{{year}}`

**Capability #2:**
- Tell me the the number of reviews that our store received by far that mention term "`{{term}}`"

**Capability #3:**
- What brands appear most frequently among the top search terms?
- List the top `{{n}}` search terms in my store

**Capability #4:**
- Telll me the grand total of invoice `{{id}}`.

**Capability #5:**
- Presents the monthly count of successful orders `{{period}}` in MM:COUNT format

**Capability #6:**
- What's the total number of items sold in the most recent `{{k}}` orders?

**Capability #7:**
- Show all customers

**Capability #8:**
- Give me the `{{Attribute}}` of the products that have `{{N}}` units left

**Capability #9:**
- Get the total payment amount of the last `{{N}}` `{{status}}` orders

**Capability #10:**
- Find the customer name and email with phone number `{{PhoneNum}}`

**Capability #11:**
- Tell me the `{{attribute}}` of the customer who has the most cancellations in the history
- Which customer has completed the `{{quantifier}}` number of orders in the entire history?
- Show me the `{{information}}` of the customer who is the most unhappy with `{{product}}`

**Capability #12:**
- How many reviews our shop received `{{time}}`?
- What is the total count of `{{status}}` reviews amongst all the reviews?

**Capability #13:**
- Preview the `{{name}}` theme for my shop

**Capability #14:**
- Mark all `{{brand}}` shirts on sale

**Capability #15:**
- Disable `{{product}}` from the site, they are facing some quality issues.

**Capability #16:**
- `{{action}}` the price of `{{config}}` by `{{amount}}`
- `{{action}}` the price of this product by `{{amount}}`

**Capability #17:**
- Update the description of `{{product}}` to highlight the real user positive reviews by quoting the comments

**Capability #18:**
- Cancel order `{{id}}`

**Capability #19:**
- Change the page title of "`{{old-heading}}`" page on my site to "`{{heading}}`".

**Capability #20:**
- Notify `{{name}}` in their most recent pending order with message "`{{message}}`"

**Capability #21:**
- Update order #`{{order}}` with the `{{service}}` tracking number `{{tracking}}`

**Capability #22:**
- Make all `{{product}}` as out of stock

**Capability #23:**
- Modify the address of order #`{{order_id}}` to `{{address}}`

**Capability #24:**
- Add new `{{option}}` `{{value}}` to `{{base_setting}}` of `{{product}}`

**Capability #25:**
- Lookup orders that are `{{status}}`
- Get the `{{attribute}}` of the `{{status}}` order

**Capability #26:**
- Add a simple product named `{{product}}` with `{{stock}}` in stock, available in size `{{size}}` and color `{{color}}`, priced at $`{{price}}`

**Capability #27:**
- Draft a new marketing price rule for `{{topic}}` that offers `{{rule}}` for all customers

**Capability #28:**
- Today is 3/15/2023, generate a `{{report}}` `{{time_span}}`
- Create a `{{type}}` report from `{{start_date}}` to `{{end_date}}`

**Capability #29:**
- We've received `{{quantity}}`, update the inventory.

**Capability #30:**
- Approve the positive reviews to display in our store.

**Capability #31:**
- Delete all `{{review_type}}`

**Capability #32:**
- Tell me the full address of all `{{airport_type}}` that are within a driving distance of `{{radius}}` to `{{start}}`

**Capability #33:**
- What is the `{{information}}` of `{{location}}`
- I will arrive `{{place}}` soon. Provide the name of a `{{target1}}` in the vicinity, if available. Then, tell me the `{{information}}` to `{{target2}}` from the hotel.

**Capability #34:**
- What is the zip code of `{{place}}`?

**Capability #35:**
- Given the following locations, `{{place_list}}`, what would be the optimal route to travel through them all in order to minimize total travel time? Please note the journey begins at the first place listed.

**Capability #36:**
- Which US states border `{{state}}`?

**Capability #37:**
- Where is the nearest `{{places}}` to `{{start}}`, and what is the walking distance to it?
- Find the walkway to the closest `{{store}}` from `{{location}}`.
- How long does it take to walk from `{{start}}` to `{{end}}`?
- Tell me the closest `{{place1}}`(s) to `{{place2}}`

**Capability #38:**
- From my stay at `{{hotel}}`, what's the estimated driving time to reach `{{place}}`?
- What is the minimum travel time by car from `{{location1}}` to `{{location2}}`?
- What is the duration required to first walk from `{{place_A}}` to `{{place_B}}`, and then drive to `{{place_C}}`?
- Show me the walking distance from nearby hotels to `{{location}}` that take at most `{{n}}` minutes?
- What is the estimated driving time between `{{city1}}` and `{{city2}}`?

**Capability #39:**

- From my stay at `{{hotel}}`, what's the estimated driving time to reach `{{place}}`?
- What is the estimated driving time between `{{city1}}` and `{{city2}}`?
- I am at CMU Pittsburgh, how long it takes to drive to the nearest `{{location}}`
- Check if the `{{place}}` in pittsburgh can be reached in one hour by car from `{{location}}`

**Capability #40:**

- Find the `{{space}}` around `{{location}}`
- Find the walkway to the closest `{{store}}` from `{{location}}`.
- Tell me the closest `{{place1}}`(s) to `{{place2}}`
- Where is the nearest `{{location}}` from `{{location2}}` `{{condition}}`

**Capability #41:**

- What is the `{{information}}` of `{{location}}`
- Tell me the coordinates of `{{location}}` in DD format

**Capability #42:**

- How much time does it take from Pittsburgh to Philadelphia by car?

**Capability #43:**

- Show the route from SCS CMU in Pittsburgh to the location where the Declaration of Independence and Constitution were signed

**Capability #44:**

- Pull up the description page of `{{location}}` on Map
- What is the `{{information}}` of `{{location}}`

**Capability #45:**

- I am arriving at Carnegie Mellon University. Find the nearby US Citizenship and Immigration Services and the walking distance to the nearest Social Security Administration from US Citizenship and Immigration Services

**Capability #46:**

- I am arriving at Pittsburgh Airport. Show me the name of a Hyatt hotel if there is any nearby. Tell me the names of supermarkets that are within 15mins driving from the hotel

**Capability #47:**

- Measure distance between `{{location/address_1}}` and `{{location/address_2}}` by walking
- Get directions from `{{location/address_1}}` to `{{location/address_2}}` using `{{transportation}}` options.

**Capability #48:**

- List out reviewers, if exist, who mention about `{{description}}`

**Capability #49:**

- Today is 6/12/2023. Tell me how many fulfilled orders I have `{{period}}`, and the total amount of money I spent.

**Capability #50:**

- Tell me the status of my latest order and when will it arrive

**Capability #51:**

- What is the date when I made my first purchase on this site?

**Capability #52:**

- I have jaw bruxism problem, show me something that could alleviate the problem.

**Capability #53:**

- What is the price range for products from `{{brand}}`?
- What is the price range of `{{product}}` in the One Stop Market?

**Capability #54:**

- How much I spent on `{{category}}` shopping during `{{time}}`

**Capability #55:**

- What is the `{{option}}` configuration of the `{{product}}` I bought `{{time}}`
- I previously ordered some `{{product}}` `{{time}}` and later cancelled. Can you reorder it for me?

**Capability #56:**

- I have a lot of Nintendo Switch game cards now, help me find the best storage option to fit all `{{num}}` cards

**Capability #57:**

- What are the main criticisms of this product? Please extract the relevant sentences.

**Capability #58:**

- What do customers say about `{{product_type}}` from `{{manufature}}`

**Capability #59:**

- Buy the best rating product from "`{{category}}`" category with at least 5 reviews and the product is least expensive
- I am doing a market survey for one stop market, show me the most expensive product from `{{product_category}}` category
- Buy the highest rated product from the `{{product_category}}` category within a budget `{{dollar_value}}`.

**Capability #60:**

- Search for "`{{keyword}}`"

**Capability #61:**

- List the full product names of slide slippers from Nike and tell me the price range of the available products

**Capability #62:**

- Look up the most recent models of XBox controllers released between 2020-2021?

**Capability #63:**

- Show the least expensive `{{product}}` with a minimum storage capacity of `{{min_storage}}`.

**Capability #64:**

- Show the most recent `{{status}}` order
- Get the order number of my most recent `{{status}}` order

**Capability #65:**

- Which number to call for the customer service?

**Capability #66:**

- How much refund I should expect from my order canlled in `{{time}}`? I only kept the AC-DC Adapter and the shop told me that I cannot get the shipping fee back

**Capability #67:**

- Show me the "`{{product}}`" listings by `{{sorting_order}}`.

**Capability #68:**

- How much did I spend on shopping at One Stop Market `{{time}}`? They gave me a 20% discount on the total amount for orders exceeding $200 in cash

**Capability #69:**

- Tell me when I last ordered my `{{description}}`?

**Capability #70:**

- List products from `{{product_category}}` category by `{{order}}` price
- Show me products under $`{{price}}` in "`{{product_category}}`" category

**Capability #71:**

- Show me the `{{info}}` for order number `{{order_number}}`.

**Capability #72:**

- find discounted items.

**Capability #73:**

- Summarize customer reviews for `{{product}}`.

**Capability #74:**

- List the customer names who thinks EYZUTAK phone cases are of good looking
- Who gave `{{stars}}` for phone cases from EYZUTAK

**Capability #75:**

- What is the rating of `{{product}}`

**Capability #76:**

- Add the product with the lowest per unit price from my open tabs to the shopping cart

**Capability #77:**

- Add `{{product}}` to my wish list
- Add this product to my wishlist
- Add a `{{product}}` to my wish list.

**Capability #78:**

- Subscribe to the newsletter of OneStopMarket

**Capability #79:**

- I recently moved, my address is `{{address}}`, update my information on OneStopShopping accordingly
- Change the delivery address for my most recent order to `{{address}}`.

**Capability #80:**

- Rate my recent purchase of `{{product}}` with `{{num_star}}` stars, using my nickname `{{nickname}}`?

**Capability #81:**

- Fill the "contact us" form in the site for a refund on the {{product}} I bought, stating that it broke after just three days of use. Also, ensure to include the order number #{{order_id}} and the product SKU. Don't submit yet, I will check.
- Draft a refund message via their "contact us" form for the {{product}} I bought {{time}}. It broke after three days of use. The shop requires the order id, the reason and the amount to refund in the message. Don't submit yet

**Capability #82:**

- Draft an email to the shop owner via their contact us function for a coupon as {{reason}}

**Capability #83:**

- Tell me the count of comments that have received more downvotes than upvotes for the user who made the latest post on the {{forum}} forum.

**Capability #84:**

- Among the top {{number}} post in "{{subreddit}}" forum, {{description}}

**Capability #85:**

- Change my reddit bio to "{{content}}"

**Capability #86:**

- Reply to {{position_description}} with my comment "{{content_description}}"

**Capability #87:**

- Create a new forum named {{name}}, with a description of {{description}}, and include {{sidebar_list}} in the sidebar?

**Capability #88:**

- Open the thread of a trending post on the forum "{{subreddit}}" and subscribe.
- Upvote the newest post in {{subreddit}} subreddit

**Capability #89:**

- Create a discussion post about "{{topic}}" in a relevant subreddit and ask users for their opinions with the simple prompt, "your opinion"
- Find a subreddit focused on topics related to {{topic}}, and post my question, "{{question}}" there
- Post my question, "{{question}}" in a subreddit where I'm likely to get an answer

**Capability #90:**

- Post a review of my recent reading "{{book}}" in the r/books with my comment "{{content}}".

**Capability #91:**

- Re-post the image of {{content}} in this page to {{subreddit}} subreddit and note "from /f/pics"

**Capability #92:**

- Ask for advice about {{issue}} in a subreddit for relations

**Capability #93:**

- Post in the most appropriate subreddit and ask for recommendations for {{category}} products within a budget of {{price}}
- Ask for product recommendations for {{category}} within a budget of {{price}} in {{subreddit}}

**Capability #94:**

- Post a notice on a virtual meetup for {{interest}} enthusiasts on {{date}} in the {{subreddit}} subreddit

**Capability #95:**

- Post in {{subreddit}} subreddit about what could diffusion model help the correpong field.

**Capability #96:**

- Thumbs down the top {{k}} post ever in {{subreddit}}.

**Capability #97:**

- Like all submissions created by {{user}} in subreddit {{subreddit}}
- DisLike all submissions created by {{user}} in subreddit {{subreddit}}

**Capability #98:**

- Edit my post on {{post}} by adding a line to the body that says "{{content}}"

**Capability #99:**

- Check out my todos

**Capability #100:**

- Check out the most recent open issues

**Capability #101:**

- Checkout merge requests assigned to me
- Checkout merge requests requiring my review

**Capability #102:**

- Tell me the full names of the repositories where I made contributions and they got {{description}} stars?

**Capability #103:**

- Open my latest created issue that has {{keyword}} in its title to check if it is closed
- Open my latest updated issue that has keyword "{{keyword}}" in its title to check if it is closed

**Capability #104:**

- See all public projects

**Capability #105:**

- Get me my RSS feed token

**Capability #106:**

- Show me the command to clone {{repo}} with SSH.

**Capability #107:**

- List all opened issues {{description}}

**Capability #108:**

- Who else have access to my repo {{repo}}, show me their usernames

**Capability #109:**

- Post "{{content}}" for the merge request related to {{mr}} in {{repo}} project

**Capability #110:**

- Fork {{repo}}.

**Capability #111:**

- Make the LICENSE of {{repo}} to MIT license.

**Capability #112:**

- Go to the merge request on {{topic}} I have to review, find if the author of the merge request responded at the end, and reply "Thank you" if he did. Otherwise remind him with a simple @.

**Capability #113:**

- Set my gitlab status as {{status}}.

**Capability #114:**

- Update the project site's title to "{{title}}"

**Capability #115:**

- set the homepage URL on my GitLab profile to {{url}}

**Capability #116:**

- Set up a new, empty repository with the name {{project_name}}?
- Create a private {{template}} repository called "{{project_name}}" using the right template to speed up development.

**Capability #117:**

- Invite {{collaborator_account_list}} as collaborator to {{repo}} repo
- Add the following users to repo {{repo}} as {{role}}: {{user_list}}

**Capability #118:**

- {{name}} wants to check my dotfile configurations. Please invite him to the repo as a guest.

**Capability #119:**

- Star the top {{number}} most stared repos in Gitlab

**Capability #120:**

- Follow {{account_list}} on Gitlab

**Capability #121:**

- Create a milestone for the upcoming {{event}} starting on {{start_date}} and ending on {{end_date}}

**Capability #122:**

- Create an issue {{issue}} in {{repo}}.
- Assign the issue regarding {{issue}} in {{repo}} to {{account}}.
- Create an issue in {{repo}} repo with title "{{issue}}". Assign the issue to {{account}}. Set due date to be {{due}}

**Capability #123:**

- Submit a merge request for `{{source_branch}}` branch to be merged into `{{target_branch}}` branch, assign `{{reviewer}}` as the reviewer

**Capability #124:**

- Open a new issue to discuss the implementation of `{{feature}}`

**Capability #125:**

- Start a private project `{{project_name}}` with `{{template}}` template and add `{{account_list}}` as members

**Capability #126:**

- How many commits did `{{user}}` make on `{{date}}` in total?
- Tell me who has made the most contributions, in terms of number of commits, to the `{{repo}}` project
- Tell me the `{{attribute}}` of the contributor who has the most commits to branch `{{branch_name}}`
- List the `{{attribute}}` of the top 3 contributors to `{{repo}}` repo, ranked by the number of commits?

**Capability #127:**

- create a new group "`{{name}}`" with members `{{members}}`

**Capability #128:**

- Tell me the distance to drive from Carnegie Mellon University to the top computer science school in massachusetts

**Capability #129:**

- What's the closest national park to `{{city}}`? How long does it take to bike there?

**Capability #130:**

- Find the page of `{{description}}` on the map.

**Capability #131:**

- Show me the way from `{{location}}` to the home stadium of `{{sport_team}}` `{{time}}`

**Capability #132:**

- Find a GitLab repository related to `{{topic}}` and make a Reddit post linking to it in a relevant subreddit
- create a repository named `{{name}}` that includes a README file with the links to the most active `{{num}}` DIY ideas on DIY subreddit?
- Make a folder named `{{directory}}` on the `{{gitlab_repo}}` repo and include a file called urls.txt that consists of the links to the 5 most recent posts from `{{subreddit}}`.

**Capability #133:**

- Promote `{{repo}}` to subreddit `{{subreddit}}` with the description from the repo itself.

**Capability #134:**

- Create a repo named `{{name}}` with `{{topics}}` in a README file

**Capability #135:**

- Gather the titles of `{{product}}` reviews with `{{rating}}` rating from OneStopShop, and post them in the games subreddit under the title "real user feedback on `{{product}}`"

**Capability #136:**

- Show me the route and driving time from `{{city1}}` to `{{city2}}`

## G PROMPTS

We provide the prompts used to generate novel out-of-domain objectives, urls, web pages, and solution trajectories.

**Generate Novel Synthetic Objectives and Websites:**

```
Here are a few example objectives (tasks) a user might be asked to perform on a webpage.
Closely following these example objectives, generate a potential objective a user might
want to perform on another American website that is similar to the examples.  (in terms of
reasoning required, requiring navigating to multiple pages or taking multiple steps to solve,
etc.)  The new objective should not be on a website that is the same or is similar to any of
the example objective's websites/domains, it should be a completely different website.  Ensure
the objective has a definitive, objective answer, and not a subjective answer.  Return just
the objective and a domain name (no path in the URL, just the hostname) of the website (in the
same OBJECTIVE:/URL: format) and nothing else.

OBJECTIVE: {...}
URL: {...}

{...other examples}
```

**Generate Plan for Hypothetical Synthetic Solution Trajectory:**

```
OBJECTIVE: {...}
URL: {...}

Here is an objective a user can perform on the webpage.  The user may need to perform multiple
actions / steps (clicking, typing, scrolling, storing/remembering information, or recalling
stored information) in order to solve the objective.  Assuming the user is starting with a
web browser that is already loaded with the website, output the required / necessary steps the
user must take on the page to solve the objective, one step per line.  Each step MUST involve
either clicking, scrolling, typing, or stopping (when the objective is complete).  DO NOT
output steps that don't involve one of these actions.  If a step does not involve clicking,
scrolling, typing, or stopping, such as remembering/recalling/calculating information, combine
it instead with the next step in the sequence that does.  Return nothing else other than the
necessary steps, no bullets and no numbered lists.
```

**Generate Hypothetical URL for Random Step in Synthetic Trajectory:**

```
OBJECTIVE: {...}
WEBSITE: {...}
STEPS:
1.  {...}
2.  {...}
{...other steps}

Here is an objective a user can perform on a website starting from the homepage and some
steps a user may take to solve the objective.  Output a realistic and valid URL (don't use
placeholders like '123', 'example', 'acme', etc.)  for what page a user would be on after they
perform Step #{...}.  Return just the URL and nothing else.
```

**Generate Hypothetical Previous and Next Action for Random Step in Synthetic Trajectory:**

```
Here are 2 example objectives a user might be asked to perform on a URL / webpage (provided in
accessibility tree format).  The goal is to perform a series of incremental actions that can
complete the objective.  The previous action that was taken and the next action a user should
take towards completing the objective along with a "Let's think step-by-step." explanation is
also provided for the 2 examples.  All actions possible for the user are:

{...}

The action should always be placed inside ``````.  For example, "In summary, the next action I
will perform is ```click [1234]```".

Example 1:

OBJECTIVE: {...}
URL: {...}
WEBPAGE: {...}
PREVIOUS ACTION: {...}
NEXT ACTION: {...}

Example 2:

{...other example}

Following the structure of these 2 examples closely, for the objective and URL below, generate
a realistic full-length webpage accessibility tree, realistic previous action, and realistic
next action that a user needs to perform on the webpage in order to complete Step #{...} of
the OVERALL PLAN towards the objective.  Provide the actions and webpage in the same format
(WEBPAGE:/PREVIOUS ACTION:/NEXT ACTION:).  Ensure the next action is Step #{...}, the next
action begins with "Let's think step-by-step." and ends with "In summary, the next action I
will perform is ```...```", and the [id] for any actions is an ID number not a string.  Do not
mention or reference the OVERALL PLAN or Step #{...} directly in the output.  Return nothing
else other than the two actions and the webpage.

OBJECTIVE: {...}
URL: {...}
OVERALL PLAN:
1.  {...}
2.  {...}
{...other steps}
CURRENT STEP: {...}
```

**Generate Hypothetical Web Page for Random Step in Synthetic Trajectory:**

Here are 2 example objectives a user might be asked to perform on a URL / webpage (provided in
accessibility tree format).  The goal is to perform a series of incremental actions that can
complete the objective.  The previous action that was taken and the next action a user should
take towards completing the objective along with a "Let's think step-by-step." explanation is
also provided for the 2 examples.  All actions possible for the user are:

{...}

The action should always be placed inside ``````.  For example, "In summary, the next action I
will perform is ```click [1234]```".

Example 1:

OBJECTIVE: {...}
URL: {...}
WEBPAGE: {...}
PREVIOUS ACTION: {...}
NEXT ACTION: {...}

Example 2:

{...other example}

Following the structure of these 2 examples closely, for the objective, URL, previous action,
and next action below, generate a realistic full-length webpage accessibility tree (don't
use placeholders like '123', 'example', 'acme', etc.).  Ensure the page is in English and is
structured such that performing the next action described would realistically complete or make
incremental progress towards completing the objective.  Provide the webpage in the same format
(WEBPAGE:) and return nothing else other than the webpage.

OBJECTIVE: {...}
URL: {...}
PREVIOUS ACTION: {...}
NEXT ACTION: {...}

## H RESOURCES

We provide links and citations to resources used in this paper which provide license information, documentation, and their intended use. Our usage follows the intended usage of all resources.

We utilize the following models:

- GPT-4 (OpenAI et al., 2024)
- Qwen-1.5-72B-Chat (Bai et al., 2023)
- `sentence-transformers/all-distilroberta-v1` (Sanh et al., 2019; Liu et al., 2019)
- `sentence-transformers/all-mpnet-base-v2` (Song et al., 2020)

We utilize the following datasets:

- WebArena Benchmark (Zhou et al., 2023)

We utilize the following software:

- DataDreamer (Patel et al., 2024)
- PyTorch and PyTorch FSDP (Paszke et al., 2019; Zhao et al., 2023)
- QLora (Dettmers et al., 2023)
- Transformers (Wolf et al., 2019)
- Sentence-Transformers (Reimers and Gurevych, 2019)
- SymbolicAI (Dinu et al., 2024)
- fastdtw (slaypni, 2017; Salvador and Chan, 2004)

We estimate the total compute budget and detail computing infrastructure used to run the computational experiments found in this paper below:

- 4x NVIDIA RTX A6000 / 300GB RAM / 50x CPU – 900 hours

