# OpenReview forum: "Large Language Models Can Self-Improve At Web Agent Tasks"
_ICLR.cc/2025/Conference — Submitted to ICLR 2025_

### Official Review · Reviewer_1yvG · 2024-10-26

**Soundness:** 2
**Presentation:** 3
**Contribution:** 2
**Rating:** 6
**Confidence:** 3

**Summary:**

The authors study whether self-refinement (i.e. finetuning on self-generated data) can improve the performance of large language models on web agent tasks via the WebArena benchmark. This benchmark tests agents on a set of web page decision-making tasks. In such setting, self-refinement data consists of trajectories of interleaved environment observations and LM agent actions. The authors test three different ways of generating  trajectories for self-refinement: (A) fully “in-domain,” using train set tasks from the Web Arena environment only; (B) on a “in-domain” and “out-of-domain”  mixture, using train set tasks from WebArena as well as synthetic tasks generated by the LM agent; (C) fully “out-of-domain,” using only synthetic tasks. All “in-domain” trajectories are filtered using self-critique, for LM agent generation errors, and environment errors. The authors finetune a single LLM on each of these three data mixtures and compare the resultant LLM agents’ performance on WebArena under three different metrics. They show that self-refining on data mixtures A and B result in some improvements to performance on 3/3 metrics, while finetuning on mixture C produces a slight drop in performance on 2/3 metrics. Finally, the authors show that running an additional round of self-refinement with the best performing data generation strategy (mixture A) causes a drop in performance on 3/3 metrics, compared to results after a single round.

**Strengths:**

- The study of generalizable techniques for improving LLM agents is of interest to the community. While many previous works have studied the advantages and limitations of self-refinement for reasoning and simple decision-making tasks, there has been limited work on harder benchmarks with longer decision-making horizons, like Web Arena. These tasks represent the frontier of state-of-the-art LLM capabilities. As a result, developing more effective methods for self-refinement on such problems is especially important.
- The paper is polished, clear, and fairly well written. The authors expound on all experimental details.
- The default metric for LLM performance on WebArena is the average completion rate of environment tasks. The authors make an effort to address the lack of granularity in this metric by introducing two auxiliary metrics for assessing LLM agentic performance. These auxiliary metrics add some nuance to the analysis, though this could be strengthened by increasing the number of experiment replicates and/or increasing the inference-time sample size, as models are evaluated at high sampling temperature.

**Weaknesses:**

While the questions posed by this paper are timely, the experimental results have limited scope and are much more mixed than the text would suggest. While the title reads “LLMs can self-improve at Web Agent tasks,” the results seem to suggest that the performance margins are extremely limited.

- Throughout the paper, the authors cite a 31% improvement over the base LLM agent as evidence for the power of self-improvement. However, this 31% is a relative score, corresponding to a mere ~2.2%  improvement in raw score from a completion rate of 7.14/100 to 9.36/100. The statistical significance of this improvement is unclear for two reasons: (1) the scale of standard error over 0-1 task completion score is omitted from Table 3; (2) according to Table 5, the authors sample from models during inference with a high temperature (T = 1). Sampling rather than using greedy decoding introduces an additional source of statistical error to results, which is similarly not quantified.
- In the introduction, the authors write: “We analyze three synthetic training data mixtures and find all three mixtures improve performance” (Lines 101-102).  This claim is misleading, since as shown in Table 3, self-refining on the third data mixture (C, “out-of-domain” tasks only) produces decreases in performance on 2/3 of the studied metrics.
- The authors show that iterative self-refinement produces a drop on all 3/3 Web Arena LLM agent performance metrics, even after the careful filtering procedures (Table 3, last row). As a reader, this result powerfully contradicts the tone of the title.
- On lines 404-406, the authors write: “Interestingly, we find that the majority of capabilities acquired by M_A and M_C are mutually exclusive, suggesting in-domain synthetic examples and out-of-domain synthetic examples improve acquisition of different capabilities.” This point would be extremely interesting if strengthened, but does not seem to be rigorously supported by the results. Currently, it seems to be being made on the basis of a single random seed of self-refinement in each setting, i.e. using a single sampled set of high temperature synthetic task generation + agent trajectories + LLM fine-tuning. In my view, to confirm these results, evaluation should be conducted with at minimum ~3-5 seeds in each data mixture setting.
- Overall the experiments have limited scope. Evaluations are conducted with a single seed for each data generation variant and a single LLM.

Though the results of this paper are mostly negative, I still believe they are of interest to the community. I would be willing to increase my score if (1) the authors can provide an estimate of statistical error for all reported scores; and (2) the tone of the paper is adjusted to reflect the strength and scope of the experimental results.

**Questions:**

There are a few other points that I would appreciate if the authors could clarify.

**(1)** The authors claim that the procedure for filtering in-domain self-refinement data is entirely unsupervised, but then seem to contradict this point later on.

In Lines 135-136: “Similar to the self-improvement prior work we discuss earlier, the collection of this data is completely unsupervised and no ground-truth labels are utilized for filtering and selection.”

Then, in Lines 194-196: “Finally, we also filter out any trajectories where the WebArena environment encountered an error or the model failed to produce a valid, parsable generation”

If I understand the procedure correctly, the highlighted parts of the filtering process are not truly unsupervised — they leverage signals that are external to the model, e.g. feedback from the environment. If so, the first claim should be adjusted accordingly.

**(2)** On lines 290-292, the authors write: “We note, however, that a number of tasks in the WebArena benchmark are trivial tasks and can be solved by a trivial baseline agent or weak model that performs no actions and only immediately exits by always generating stop [N/A].”

This seems like a major weakness of the benchmark. What proportion of the WebArena tasks have this degeneracy property? How many of the <10% of tasks solved by tested LLM agents are degenerate? From my understanding of the text, degenerate tasks **are** filtered from capability score computation, but **are not** filtered from/identified during evaluation. Is this correct?

---

> ### Author Response · Authors · 2024-11-25
>
> We thank the reviewer for their detailed and constructive feedback, as well as their recognition of the importance of generalizable self-refinement techniques, the novelty of the metrics proposed, and the clarity of the paper. We thank the reviewer for their willingness to reconsider the score following our responses and for their valuable insights that will help improve the paper. We address the key concerns raised below.
>
> **1. Statistical Error**
> In our revision PDF, we now also provide confidence intervals and highlight significance. We also provide more experiments of our technique on Llama-3-70B, showing the procedure generalizes to different models.
>
> **2. Relative vs. Absolute Performance**
> While the absolute performance is small, the relative performance improvement (31%), is large. We believe this performance improvement is quite significant as WebArena is an extremely challenging task with even frontier models like GPT-4 performing extremely poorly on the task with only ~14% functional correctness. Therefore, absolute performance scores would be expected to remain low, and relative improvement can represent a significant gain in model ability. We address revisions to tone and claims in the next two points where claims could potentially be interpreted by readers as overstated.
>
> **3. Mixture C Performance Claim**
> “We analyze three synthetic training data mixtures and find all three mixtures improve performance” - We find all 3 mixtures do improve performance on capability score, which is the strongest measure of agent performance, since it ignores degenerate tasks. We understand the reviewer’s concern that this claim could be misleading and have adjusted the claim in the revision to claim all mixtures improve capabilities (as measured by the capability score), instead of general performance. We believe this addresses the concern, and we are happy to further adjust the claim and tone at the direction of the reviewer.
>
> **4. Iterative Self-Improvement / Tone**
> As discussed in our paper, works like Chen et al. [2024] and Feng et al. [2024] also observe diminishing returns in self-improvement, suggesting this is a general limitation of self-improvement techniques rather than a specific one to our work. We appreciate the reviewer’s concern that this might be confusing to some readers, given the tone, and we have clarified in the revision, in the introduction, that while models can self-improve, our results do *not* show that continual, iterative self-improvement is possible. We believe this addresses the concern, and we are happy to further adjust the claim and tone at the direction of the reviewer.
>
> **5. Limited Scope**
> Web agent benchmarks like WebArena are computationally intensive due to latency from real-time web interactions, where LLM actions can't be batched. Fine-tuning large models with 70B+ parameters is also demanding, therefore we had to choose a singular research question to explore. We agree with the reviewer there are many other angles yet to explore and the results can be strengthened through further repeated trials and hyperparameter search, but thank the reviewer for also acknowledging the current contributions will be of interest to the community.
>
> **6. Clarifications**
> We appreciate the reviewer’s requests for clarification and address them below:
> - **Unsupervised Filtering**: While the filtering process does not rely on ground-truth labels, it does leverage external signals from the environment (e.g., errors, invalid generations). We will revise the text to clarify that this process is unsupervised in the sense that it does not require labeled data but does depend on environment-provided feedback
> - **Degenerate Tasks in WebArena**: We find approximately 38 tasks out of 812 tasks are degenerate (4.68%). To clarify, degenerate tasks are excluded from the capability score computation as we require an agent must complete at least one non-degenerate task belonging to a capability. We don’t filter them from the “functional correctness” score computation as this is the score used by the WebArena paper, which computes this score with all 812 tasks, including the degenerate ones. We compute this score as is, as provided by the scripts provided by the WebArena framework. We propose the “capability score” as an alternative measure that we believe is a better measure of an agent’s skill display and we find that the agent improves on this metric in all 3 mixtures.

---

### Official Review · Reviewer_neMT · 2024-10-29

**Soundness:** 1
**Presentation:** 2
**Contribution:** 2
**Rating:** 3
**Confidence:** 5

**Summary:**

This paper proposes a method of improving LLMs for web agent tasks. The method involves prompting an existing LLM as an agent to tackle tasks in WebArena. The trajectories are then filtered down to retain those that are likely to be successful. The model is then finetuned (with QLoRA) on these trajectories. In the paper, alternative trajectory generation methods are also explored, such as using the successful WebArena trajectories to generate out-of-domain trajectories on other websites. In addition, the authors propose a new metric for measuring the similarity of an agent trajectory, which has benefits over using the functional correctness evaluation in WebArena.

**Strengths:**

- The paper shows that you can use in-domain trajectories as few-shot examples to generate novel out-of-domain trajectories, which will be useful in producing synthetic data for finetuning agents.
- The paper proposes a new metric $\text{VERTEX}_{\text{DTW}}$ which measures the similarity of an agent’s trajectory w.r.t. a reference (in this paper, the reference is a gpt-4 trajectory). This can be useful for providing a soft metric that has less false negatives than the WebArena functional correctness evaluation.

**Weaknesses:**

My primary concern with this paper is that its evaluation setup is not fair. The models are essentially *training on the test set*, and doing a pass@2 on the evaluation set (these results align with the ablation experiments in the paper, where training on Mixture C actually makes the agent worse). Consider the filtered subset of in-domain examples (mixture A). It has a very high accuracy rate (0.919), which is expected as tasks that errored out or the model failed to solve are filtered out. In the WebArena benchmark, models often only stop when they are successful, so the false negative rate is low. If you view mixture A as groundtruth samples, finetuning on this subset is essentially allowing the model to memorize the examples that have passed, while ignoring those that it failed on the first attempt. When you test such a model, this can be viewed as essentially taking a second sample for the failed tasks. This is not a fair comparison to baseline models: a fairer comparison would be the same baseline model but an accuracy score of its pass@2 (i.e., take the best run out of two samples using the same filtering criteria). [1] actually does this with a prompted value function and gets a similar relative improvement (29%) compared to this work, and they do not do any training on the samples. If the authors can (1) compare their results against pass@2 of the agent (while resetting the WebArena environment after the first run, to avoid bias), and (2) perhaps run their model on a different benchmark, e.g., Mind2Web, VisualWebArena, WorkArena, or WebLINX, to show generalization, this would alleviate some of my concerns.

In addition, it would also be good to have error bars on the evaluation results as WebArena can be noisy, as the absolute difference between the baseline and the self-improved agent is low (7.14 to 9.36) and could be within the error range.

The proposed $\text{VERTEX}_{\text{DTW}}$ metric is evaluated against gpt-4 generated outputs, which possibly introduces model bias in the evaluation. WebArena has [human demonstrations](https://github.com/web-arena-x/webarena/blob/main/resources/README.md#12212023-human-trajectories), and it would be good to compare against these instead.


Overall I think the evaluation of the paper can be greatly improved (by measuring on more benchmarks, to avoid train-test contamination), as well as more ablation experiments. For these reasons I don’t think that the paper in its current shape is sufficiently complete.

**Questions:**

How many examples are in Mixture C? How does the result scale with the number of examples used in Mixture C? Does it improve as more examples are provided?

---

> ### Author Response · Authors · 2024-11-25
>
> We thank the reviewer for their thorough and constructive feedback. Below, we address the primary concerns raised and clarify misunderstandings:
>
> **1. Evaluation Setup and Fairness**
> We acknowledge the reviewer’s concerns regarding the evaluation setup and believe we can thoroughly address each concern. Specifically:
>
>
> - **Concerns of Unfair Evaluation (Mixture A)**: We believe this concern comes from a misunderstanding of the paper, and are happy to make it more clear in a revision. We would like to clarify and stress that our procedure is *fully unsupervised*. We are not using ground truth labels from the benchmark for supervised training. To collect the in-domain dataset, we run an agent on the WebArena dataset and use the agent’s *self-critiques* to filter the resulting trajectories, keeping only those the model “believes” it successfully completed. We then fine-tune on this filtered set to enable the model to self-improve by bootstrapping a fine-tuning dataset.
> - **Size of Mixture C, and Scaling Impact**: Our dataset sizes can be derived from Algorithms 1 and 2, but we will improve this for readability. For the in-domain dataset (Mixture A), we used 58 trajectories, sampling the initial action, final action, and two intermediate actions per trajectory as shown in Algorithm 1, resulting in 232 (58*4) rows of (observation, previous action, next action). The out-of-domain dataset (Mixture C) matches this size, as shown in Algorithm 2, line 4. Mixture B combines both datasets for a total of 464 rows (232*2). We kept Mixture C and Mixture A the same size for fairness and did not experiment with scaling the Mixture C dataset to different sizes.
> - **Additional benchmarks**: At the time of this study, WebArena was the most comprehensive, diverse, and challenging long-horizon benchmark available. Previous works we cite are often tested on simpler, artificial environments like ALFworld or WebShop. In contrast, WebArena is significantly more challenging, diverse, and realistic, as described by its authors. Many of these prior datasets are simulated, non-realistic environments that WebArena aims to replace. Thus, demonstrating success on WebArena offers our method the best chance for success at generalizing to real-world tasks. However, in our revision PDF, we provide more experiments of our technique on Llama-3-70B, showing the procedure generalizes to different models.
>
> **2. Error Bars and Result Robustness**
> In our revision PDF, we now also provide confidence intervals and highlight significance.
>
> **3. GPT-4 References for VERTEX_DTW Metric**
> The reviewer noted that VERTEX_DTW evaluations rely on GPT-4-generated outputs as references, which could introduce model bias. This is an excellent point, and we agree that comparisons against human demonstrations in WebArena would provide a more unbiased baseline. However, there are two issues with the human baseline provided by WebArena and why we instead choose GPT-4 as a reference for the metric:
>
> - Human traces are only available for 179 tasks while the full suite consists of 812 total tasks. Therefore, it makes the human traces available unsuitable for use to compute the VERTEX_DTW metric.
> - There is also only a single human trace available per task for WebArena. For VERTEX_DTW, we ideally have access to a distribution of possible solutions, which we approximate by sampling multiple successful trajectories from the reference models. This represents a core capability of the VERTEX score to measure capabilities when they can be solved in multiple ways.

---

> ### Comment · Reviewer_neMT · 2024-11-27
>
> Thank you to the authors for the detailed response. Let me clarify the main issue I have:
>
> > We would like to clarify and stress that our procedure is fully unsupervised.
>
> This is not true, as you are prompting the model with the same task descriptions from the WebArena test set, and filtering them with a heuristic that has very high accuracy. It is not fully unsupervised as the model is finetuned on data that contains information from the test intents. When a model successfully completes a task on WebArena, it is fairly straightforward for modern LLMs to classify it as a failure/success (see [1] for example). As mentioned in my initial review, your approach is therefore similar to doing a pass@2 of the agent, since you finetune on these trajectories and proceed to test on them again.
>
> If you can show results for the pass@2 (i.e., for every task, running the model twice with different random seeds since you are using ancestral sampling, and taking the max of both runs) of the same agent model this would be a very informative data point as well.
>
>
> Thank you for making the other changes, I think this will make the paper more readable. However I still think that the evaluation setup is unfair and the paper would benefit from another round of revisions, including measuring its performance on true held-out sets, so I am maintaining my previous recommendation. I agree with what the authors say about ALFWorld and WebShop, but there are other benchmarks now that are much more realistic. For example the [BrowserGym](https://github.com/ServiceNow/BrowserGym) framework would allow you to easily test your approach on several other realistic benchmarks (at least VisualWebArena, WorkArena, AssistantBench, and WebLINX at this point in time) to demonstrate true held-out performance.
>
>
> **References**
>
> [1] Pan, Jiayi, et al. "Autonomous evaluation and refinement of digital agents." arXiv preprint arXiv:2404.06474 (2024).

---

> > ### Author Response · Authors · 2024-11-28
> >
> > We thank the reviewer for their thoughtful response and believe we understand the concern. We think **the evaluation setup is well-accepted in the agent-learning published literature**. For example, from some of the works we cite as related work in our submission:
> >
> > *"...where the policy tries to reproduce the good behaviour demonstrated by the agent…filtering out the unsuccessful trajectories from the replay buffer and training the agent only on the high-reward trajectories"* - Gulcehre et al., 2023 (**Google Deepmind/CoRR 2023**)
> >
> > *“During the exploration phase, this base agent interacts with the target environment to undertake a set of given tasks and receive feedback from the environment. We gather failed trajectories from the base agents...in the subsequent training phase, ... to fine-tune the LLM policy with these..., thereby further improving the agent…”* - Song et al., 2024 (**ACL 2024**)
> >
> > Many of these prior works rely on a true supervised signal (like actual human supervision), rather than a heuristic/self-critique technique, which we use in this work to maintain an unsupervised setting.
> >
> > The weakness the reviewer is discussing is one that **many RL-type approaches have in general** when applied to agent environments, which is that they may overfit the environment. This general research direction/technique is still useful in practice. For example, a company deploying a web agent to perform tasks in a fixed environment of a handful of websites can use simple trajectory logging and unsupervised signals/heuristics we outline to collect “plausibly successful trajectories” to create a fine-tuning dataset to update the policy and better fit the fixed environment. This way, future agent runs capitalize on learning from successful trajectories from the initial policy. In high-demand environments, this can be very useful, as a one-time fine-tune can update the policy to better fit the environment so the agent can more efficiently and consistently perform on all future user requests. We believe our experiments demonstrate this with enough evidence to be deployed in such an industry application of agents.
> >
> > We are happy to list the general concern of generalization to other environments as a “Limitation” in our paper to acknowledge the general limitations of offline RL techniques, along the lines of: “The approach we investigate for Mixture A is a form of offline growing batch RL (Gulcehre et al., 2023) and may be subject to overfitting to the environment that trajectories are sampled from.” We believe this will satisfy the concern of this reviewer and are happy to list additional limitations at the direction of the reviewer and the AC. However, we believe the work still has strong merit and builds upon and supplements prior published RL/agent-learning works studying a similar evaluation setup.
> >
> > As for the other environments like WebLINX, VisualWebArena, and BrowserGym, we agree these are good, challenging environments similar to WebArena. However, they were released this year in 2024, after our experiments were started/completed, and adding them to our experiments now, given the compute-intensive nature of these benchmarks, would constitute a *significant experiment request* during the ICLR rebuttal period, which will unfortunately not be feasible to complete during this period.
> >
> > If this response (and our prior revisions) has satisfied or partially satisfied your concerns, we would appreciate the reviewer reflecting/updating this in the score for this submission. Thank you again for your participation in the discussion during this holiday week.

---

### Official Review · Reviewer_xuoK · 2024-11-02

**Soundness:** 3
**Presentation:** 3
**Contribution:** 2
**Rating:** 6
**Confidence:** 4

**Summary:**

solving complex, multi-step, long-horizon tasks using prompting or fine-tuning poses various challenges. The paper presents a sound approach to enhance the capabilities of LLMs through self-improvement on web automation task. They propose evaluation techniques and effective use of in-distribution and out-of-distribution synthetic data, showcasing some performance gains.

**Strengths:**

### strenghts:
1. **Capabilities through Self-Improvement:** The paper demos how LLMs can extend their capabilities through self-improvement techniques, particularly in the context of complex, long-horizon web agent tasks. This ability to acquire new capabilities while largely retaining existing ones is notable.
2. **Appropriate Eval Metrics:** The introduction of novel metrics, and scores to evaluate the quality of trajectories, adds depth to the evaluation process. These metrics provide a nuanced view of the models' performance, moving beyond simple task completion metrics and enabling a more detailed assessment of the agent's robustness and capabilities. the additions seem appropriate  enough to capture both success, relevance, and efficiency of trajectoriesl
3. **Synthetic Data Utilization:** The use of both in-domain and out-of-domain synthetic data for fine-tuning is commendable. This approach not only addresses the challenge of data scarcity but also demonstrates a systematic mechanism for enhancing agent model generalization.

**Weaknesses:**

Good to address:
1. **Hyperparameter Selection:** The paper lacks a clear justification for the choice of hyperparameters in the synthetic data generation process and generating new objectives. e.g. using 4 or 2 few-shot samples, temperature value, and perhaps just one sentence on why 0.7 cosine similarity used is better. Any missing details may lead readers to question the replicability and robustness of the results. A more thorough analysis or rationale for these choices would strengthen the paper.
2. **Limited Model / Results:** The study's reliance on a single or fewer models limits its generalizability. Expanding the experiments to include a subset of models on appropriate agent frameworks that came out recently, such as smaller or different architectures, could provide insights into how these techniques scale or vary across different model sizes and types. I am thinking aloud if the results section can be tightened and elaborated more. But I will wait to see if my other reviewers has some feedback or ideas on this front.

**Questions:**

While the study is promising, its impact would be further enhanced by addressing the concerns regarding the selection of certain variables used in the experiments and evals and model diversity.

### Suggestions for Improvement:
- **Expand Hyperparameter Exploration:** Providing a sensitivity analysis or a justification for the selection of hyperparameters used in generating synthetic data would enhance the transparency and reproducibility of the results.
- **Broader Agent Evaluation:** Including diverse model architectures in the experiments could help validate the proposed self-improvement techniques across a range of LLMs, potentially increasing the impact and applicability of the findings.

I would be happy to increase the score post author responses.

---

> ### Author Response · Authors · 2024-11-25
>
> We thank the reviewer for their thoughtful and constructive feedback, as well as their recognition of the strengths of our work. We address the key concerns and suggestions below. We thank the reviewer for their willingness to reconsider the score following our responses and for their valuable insights that will help improve the paper.
>
> **1. Broader Agent Evaluation:** To address your concerns, in our rebuttal PDF, we provide more experiments of our technique on Llama-3-70B, showing the procedure generalizes to different models. We also provide confidence intervals and highlight significance.
>
> **2. Hyperparameter Exploration:** We appreciate the comment on hyperparameters and agree hyperparameter exploration would further strengthen the work, we have added this to the “Limitations” section in our revision PDF. Web agent benchmarks like WebArena are computationally intensive due to latency from real-time web interactions, where LLM actions can't be batched. Fine-tuning large models with 70B+ parameters is also demanding, therefore we did not do significant hyperparameter exploration. We chose many of our sampling parameters (temperature and top_p) directly from choices made in the original WebArena paper. We chose 4-shot or 2-shot examples in prompts based on what was computationally feasible for our environment, although we acknowledge that these choices are arbitrary. Similarly, we chose the 0.7 cosine similarity threshold based on what was computationally feasible, as the more strict the threshold value is, the longer it may take to sample enough diverse synthetic objectives. We have now discussed this in the “Limitations” section in our revision PDF.

---

> > ### Comment · Reviewer_xuoK · 2024-11-26
> >
> > This reviewer appreciates the authors' response and updates, thank you. I will maintain my initial positive score and support for this work.

---

### Official Review · Reviewer_aP9e · 2024-11-03

**Soundness:** 2
**Presentation:** 3
**Contribution:** 1
**Rating:** 3
**Confidence:** 4

**Summary:**

The papers is about having an agent for environments with lack of training data like web browsing. In that way, LLM can be a good candidate. In this work, they introduced a new technique in which LLMs can self-improve their performance as agents in long-horizon tasks in a complex environment using the WebArena benchmark. To better understand the effect of the self-improvement method, they introduced two auxiliary metrics: 1) a measure to analyze capabilities acquired and lost by the agent and 2) an extension of the VERTEX score to measure the quality of variable-length agent trajectories.

**Strengths:**

- well written.

**Weaknesses:**

- The innovation of the work is very limited!
- Lack of enough experiments! The experiments are not comprehensive! It should consider a good analysis across different datasets and careful reasoning of what is going there! For example, having complete oblations studies versus other possibilities.
- Lack of comparison with baselines! There are a lot of works of self-improvement and self-correctness! There should be considered as baselines.
- This work needs a proper and further study and analysis! This is an incomplete work to be published!

**Questions:**

- What is the definition of plausible trajectories and high-quality?
- What are the possible oblations against the proposed type of data?

---

> ### Author Response · Authors · 2024-11-25
>
> Thank you for your detailed review and the time you dedicated to evaluating our work. We appreciate your constructive feedback and aim to address your concerns thoroughly in this response.
>
> **1. Lack of innovation**
> We outline the novelty and contributions of this work at the end of the introduction section. There has been no other work showing self-improvement on long-horizon agent tasks before. We also contribute a novel analysis of agent performance through new metrics.
>
> **2. Lack of experiments and baselines**
> WebArena is a new dataset, and at the time of writing, no other papers had implemented a self-improvement procedure on it. We show that models can self-improve by comparing their performance before fine-tuning (Base) and after (Mixture A, B, C). The evaluation we perform is consistent with the cited prior self-improvement works that have demonstrated self-improvement on far easier tasks and domains.
>
> **3. Definition of plausible trajectories**
> We provided a definition of plausible trajectories on Line 128-132: *"For all tasks in WebArena, we collect an initial set of trajectories using the base model. We filter out any trajectories where the model self-detected failure (self-critique) or failure was detectable in the environment and keep the remainder. We denote the remaining set of trajectories as plausible trajectories…we hypothesize this remaining higher-quality set of plausible trajectories can serve as reasonably high-quality in-domain synthetic examples”*

---

### Author Response · Authors · 2024-11-25
**General Response to All Reviewers**

We thank the reviewers for their positive reception and valuable feedback which helped us to substantially improve and strengthen our work. We further thank all reviewers for their consensus on novelty and impactfulness of our work on an important task that will be of interest to the ICLR community.

We are especially invigorated by the positive evaluation:

- From **reviewers xuoK, neMT, and 1yvG** for addressing the **relevance of nuanced evaluation metrics**, including auxiliary metrics and **VERTEX_DTW**, which offer more granular and robust assessments of LLM agent performance compared to standard metrics.
- From **reviewers xuoK and neMT** for emphasizing the effective use of **synthetic data**, demonstrating its value in addressing data scarcity and enhancing model generalization through both in-domain and out-of-domain trajectories.
- From **reviewers xuoK and 1yvG** for highlighting the paper’s focus on **self-improvement techniques for LLMs**, particularly in tackling **long-horizon tasks** like WebArena, a frontier of LLM capabilities.
- From **reviewers aP9e and 1yvG and xuoK** for recognizing that the paper is **well-written and polished**, with clear exposition of experimental details.

In response to the comments and suggestions, we have provided a detailed and thorough rebuttal for each reviewer, have uploaded a revised PDF with proposed revisions/additions requested by reviewers, and provide a summary of the major points for convenience:

- We noted that we did not experiment / explore different hyperparameter settings for some parameters in “Limitations”.
- We have adjusted the tone and narrowed a few sentences with claims that could be interpreted to be overstated.
- We have provided results on an **additional model**, Llama-3-70B, showing the procedure generalizes to different models.
-  We also provide **confidence intervals and highlight significance**.

We believe these significant revisions now adequately address the stated concerns of reviewers, who were willing to increase their scores, and we are happy to make further adjustments at each reviewer’s discretion in our final revision.

---

### Meta-Review · Area_Chair_ycPq · 2024-12-20

**Metareview:**

This paper performs self-improvement in LLM on the Web Arena tasks which is a challenging web navigation planning task for LLMs agents. Two main concerns were raised during the reviewing process. The first was the question of fairness in evaluation (by reviewer neMT) and the second was experimental results being weak or even misleading (reviewer 1yvG).

The fairness concern is that the agents are being trained on the same prompt/task distribution as the ones on which they are being tested. Essentially, the agent generates a set of trajectories on a given distribution of which some of the trajectories are eliminated using a self-critique signal and this data is used for training (mixture a). There is also additionally an option to generate out-of-distribution data (mixture b) and mix it with the in-distribution filtered data (mixture c). If the self-critiquing works, then it is clear that we should expect gains when training on it and then evaluating a similar distribution. A much more powerful result would have been to show that the agent can solve novel tasks after being trained. Reviewer neMT suggested doing pass@2 evaluation with baseline agents which the authors did not report. I think the comparison with RL tasks like Atari or Mujoco is tenuous. I think there is some value in overfitting to a given webpage/environment but I think the results then aren't very surprising as we know that self-critiquing helps and so training on filtered data is bound to give improvement.

The second concern is that gains are small ~2.2% absolute, using mixture C hurts, and evaluations are conducted with a single seed. Of this, authors have clarified that capability score is the most important where even with mixture C there is some gain over the base agent. However, the use of a single seed makes it harder to read a strong signal from the experiments especially when the gains are small. I do agree that training 70b models is expensive.

For these reasons, I am currently not recommending acceptance. I suggest authors to either focus on the generalization aspect and show that the agents can solve newer tasks than where they were trained. Or, add more experiments to show stronger and more reliable gains or include more domains. Ideally, both should be done. I do like the choice of WebArena and I think this paper is in an important direction.

**Additional Comments On Reviewer Discussion:**

As mentioned above, the main concerns raised by the reviewers were:

1. Fairness of experiments due to essentially training on the test set
2. Weak experiments: small gains, single seed evaluation, and arguing that a mixture of C hurts

Of the above, I think small gains could be okay provided they are reliable. And authors have explained that a mixture of C helps on the capability score which they argue is the more important.

My main concerns were, therefore, lack of generalization or surprise in results (1) and reliability of experiments (2). I think these results are addressable using the suggestions in the reviews. Finally, I discounted the review from reviewer aP9e since the review was not concrete, and the reviewer did not participate in the discussion phase.

---

### Decision · Program_Chairs · 2025-01-22

Reject